# Enhancing Numerical Prediction in LLMs via Smooth MMD Alignment

Zhuo Zuo [1]   Li Yue [2]   Wenhao Zheng [1]   Chenpeng Wang [3]   Xianggen Liu [1]

## Abstract

Despite their strong general capabilities, large language models (LLMs) often remain unreliable when outputs must be numerically precise. A key reason is the training objective: standard cross-entropy treats numeric tokens as unstructured categories and ignores the metric structure of their values. We address this mismatch with **S**mooth **M**aximum **M**ean **D**iscrepancy (**SMMD**), which builds on the classic MMD by incorporating value-distance kernels over numeric tokens and graph-based smoothness. With this kernel defined over a numeric sub-vocabulary, SMMD aligns the predicted numeric distribution to the target via kernel matching and smooths the prediction–target residual over the induced kernel graph to encourage local consistency. We evaluate SMMD on four numeric-target tasks—mathematical reasoning, arithmetic calculation, clock-time recognition, and chart question answering—across multiple open-weight LLM and VLM backbones. SMMD consistently improves accuracy over both cross-entropy and recent numeric-target losses; analyses show complementary effects between MMD and smoothness and underscore the importance of distance-based kernel design. Code is available at https://github.com/Zuozhuo/smmd-loss.

## 1. Introduction

Large language models (LLMs) have achieved remarkable progress in natural language generation and reasoning (OpenAI, 2023; Minaee et al., 2025; Guo et al., 2025), yet they remain unreliable when a task requires *precise numerical outputs* (Spithourakis & Riedel, 2018; Zausinger et al., 2025).

This weakness extends far beyond basic arithmetic word problems (Cobbe et al., 2021), manifesting in more complex *numerically grounded* contexts, from visual numerical reasoning (Masry et al., 2022; Methani et al., 2020; Kafle et al., 2018; Saxena et al., 2025) to specialized scientific and engineering workflows where precise numerical parameters directly determine the output (Zuo et al., 2025; Guo et al., 2026). In these settings, models frequently produce incorrect numerical outputs even when the surrounding reasoning appears plausible. Such failures are particularly undesirable in scientific, financial, and decision-making pipelines, where numerical errors can propagate and lead to qualitatively different outcomes.

A fundamental reason is a mismatch between the *metric structure* of numeric values and the *training signal* used to model them. In next-token prediction, numerical tokens are treated as categorical labels and optimized with cross-entropy (CE), which ignores ordinal and distance information: confusing "3" with "4" is penalized in the same way as confusing "3" with "7". Consequently, the objective provides no incentive for the model to express value proximity in its predictive distribution, even though such proximity is often crucial for numerically grounded reasoning and downstream decision making (Spithourakis & Riedel, 2018).

Recently, a growing line of work has begun to incorporate metric structure into supervision, notably through Earth Mover's Distance (EMD) (Zausinger et al., 2025; Fei et al., 2025). Concretely, EMD-based supervision penalizes the model by weighting predicted probability mass according to its distance from the ground-truth numeric token. While these principled, transport-based losses are effective, they do not explicitly encourage local smoothness in the resulting training signal. In particular, even when most probability mass is near the target, the per-token error signal can vary unevenly across neighboring numeric tokens, leaving the model's behavior less stable in the immediate vicinity of the correct value.

Motivated by these gaps, we take a kernel-distribution perspective and introduce **S**mooth **M**aximum **M**ean **D**iscrepancy (**SMMD**). SMMD adapts the classic kernel MMD framework (Gretton et al., 2012) to the discrete token distributions of LLMs and, to our best knowledge, is the first to use kernel distribution matching to supervise numeric

[1]College of Computer Science, Sichuan University, Chengdu, China [2]Dongfang Electric (Chengdu) Academy of Science and Technology Co., Ltd., Chengdu, China [3]Institute of Medical Information & Library, CAMS & PUMC, Beijing, China. Correspondence to: Xianggen Liu <liuxianggen@scu.edu.cn>.

*Proceedings of the 43rd International Conference on Machine Learning*, Seoul, South Korea. PMLR 306, 2026. Copyright 2026 by the author(s).

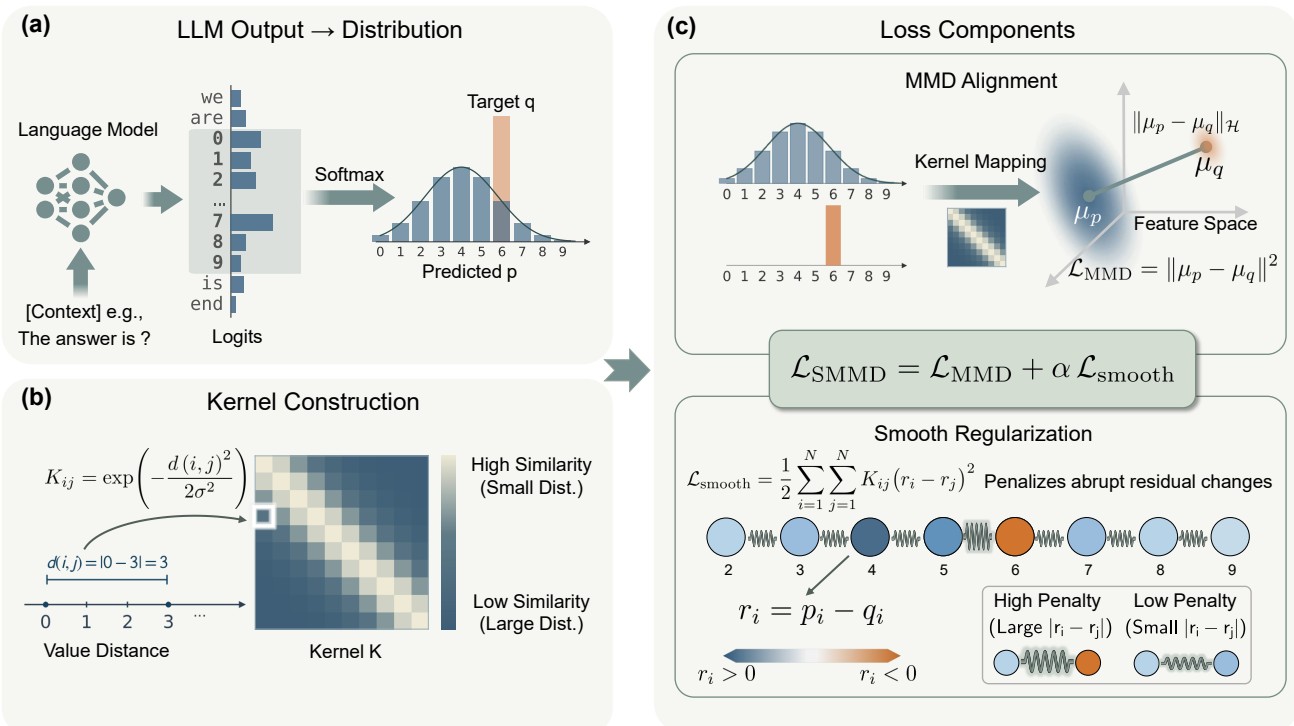

*Figure 1.* **Overview of SMMD.** **(a)** From logits, we restrict to the numeric sub-vocabulary $V_{\text{num}}$ and apply softmax to obtain the numeric distribution $p$, which is compared to the one-hot target $q$. **(b)** A value distance induced kernel $K$ is precomputed by applying a RBF kernel to pairwise gaps $|v_i - v_j|$, so numerically closer tokens have higher similarity. **(c)** Training combines kernel MMD alignment with a smoothness regularizer on the residual $r = p - q$ to encourage locally coherent errors along the numeric axis. The final objective is $\mathcal{L}_{\text{SMMD}} = \mathcal{L}_{\text{MMD}} + \alpha \mathcal{L}_{\text{Smooth}}$, where $\alpha$ is set automatically via degree-based normalization.

token prediction. Distinct from objectives that directly penalize a transport cost, SMMD takes a holistic *kernel-based* approach: it transforms value distances into a similarity kernel and aligns the predicted and target distributions by matching their moments in a Reproducing Kernel Hilbert Space (RKHS). Beyond this global alignment, SMMD further promotes *local consistency* by imposing a smoothness constraint on the prediction–target residual across nearby values via the kernel-induced Dirichlet energy. The resulting objective is lightweight, requires no architectural changes, and can be seamlessly combined with cross-entropy during training.

We evaluate SMMD on a diverse suite of numerical-output tasks spanning mathematical reasoning, arithmetic calculation, clock-time recognition, and chart question answering. Our experiments show that SMMD delivers consistent gains in numerical accuracy across a range of language-only and vision-language backbones and datasets. Analysis further indicates that the kernel matching and smoothness regularization contribute in complementary regimes, and that improvements depend on kernels that respect value-aligned distance structure. Finally, sensitivity results suggest SMMD is stable under a broad range of hyperparameter choices.

## 2. Related Work

**Numeracy in language models.** While modern LLMs excel at general-purpose tasks, their grasp of numerical values remains surprisingly brittle. Early critiques highlighted that treating numbers as standard text tokens ignores their underlying magnitude, sparking a move toward numeral-aware modeling (Spithourakis & Riedel, 2018). Much of this effort has focused on representation, ranging from digit-level tokenization to numeracy-specific training signals (Geva et al., 2020), with evidence suggesting that even subtle tokenization choices can fundamentally shift arithmetic performance (Singh & Strouse, 2024). Another direction injects more suitable inductive biases through continuous or structured encodings, especially for scientific and property-prediction settings (Golkar et al., 2024), and connects generation to continuous targets via conditional sequence regression formulations (Born & Manica, 2023). Recent representation-level work further improves single-token number embeddings through Fourier features, providing a complementary way to encode numerical structure in the model parameters (Zhou et al., 2026). Complementary work targets the sequential nature of number generation, e.g., changing digit decoding order to better align with arithmetic structure (Zhang-Li et al., 2024). Closest to our focus, several methods revise

the training signal for numeric outputs without architectural changes: NTL introduces value-aware objectives over number tokens, including Wasserstein-style loss (Zausinger et al., 2025), and NTIL further extends EMD-based supervision to encourage numerical integrity at both token and sequence levels (Fei et al., 2025). DIST[2] also injects metric distance into token-level supervision by shaping targets according to numerical proximity (Chung et al., 2026); in contrast, our SMMD keeps the one-hot target, performs kernel distribution matching in an RKHS, and further regularizes the prediction–target residual with graph smoothness. Orthogonally, inference-time strategies such as verifiers (Cobbe et al., 2021), chain-of-thought prompting (Wei et al., 2023), or program-aided execution (Gao et al., 2023) can improve accuracy, while arithmetic-oriented extended pretraining can also strengthen numeracy (Petrak et al., 2023). Overall, these threads point to a persistent objective mismatch: numeric mistakes have an inherent metric meaning (how far off the value is), but standard cross-entropy rewards only exact token matches and does not expose that structure to learning.

**Maximum Mean Discrepancy (MMD) and distribution matching.** MMD is a kernel-based distance between distributions, originally developed for two-sample testing through kernel mean embeddings (Gretton et al., 2012). In deep learning it is often used as a practical distribution-matching penalty. For domain adaptation, MMD-based losses align source and target feature distributions, with multi-kernel variants improving robustness across scales (Long et al., 2015). For generative modeling, MMD has served as a likelihood-free training signal, including adversarial variants that learn the feature space in which matching is performed (Li et al., 2015; 2017). Related kernel discrepancies also appear as regularizers for representation learning and latent-variable models when explicit likelihoods are unavailable (Tolstikhin et al., 2017; Zhao et al., 2018). Departing from the usual feature-level matching setups, we instantiate MMD as a supervised, per-token objective over a numeric sub-vocabulary, using a distance-induced kernel that reflects value proximity and pairing it with a smoothness bias to encourage locally coherent behavior along the number line.

## 3. Method

We study numeric prediction in an autoregressive language model. Given a context, the model outputs logits $\boldsymbol{\ell} \in \mathbb{R}^{|\mathcal{V}|}$ over the vocabulary $\mathcal{V}$, inducing the full next-token distribution $\tilde{\mathbf{p}} = \mathrm{softmax}(\boldsymbol{\ell})$.

The focus here is on *numeric tokens* — tokens whose string form can be deterministically parsed into a real value (via standard float casting). We precompute a numeric sub-vocabulary $\mathcal{V}_{\mathrm{num}} \subseteq \mathcal{V}$ with size $N = |\mathcal{V}_{\mathrm{num}}|$, and index it

by $\{1, \ldots, N\}$ via a bijection $\pi : \mathcal{V}_{\mathrm{num}} \to \{1, \ldots, N\}$, where index $i$ corresponds to the parsed numeric value $v_i \in \mathbb{R}$. The construction procedure is summarized in Appendix A.

At any training position whose ground-truth token $y$ lies in $\mathcal{V}_{\mathrm{num}}$, logits are restricted to $\mathcal{V}_{\mathrm{num}}$ and renormalized to form the numeric distribution:

$$\mathbf{p} = \mathrm{softmax}(\boldsymbol{\ell}[\mathcal{V}_{\mathrm{num}}]) \in \Delta^N, \tag{1a}$$

$$\mathbf{q} = \mathbf{e}_{\pi(y)} \in \Delta^N. \tag{1b}$$

Equivalently, $\mathbf{p}$ is the conditional next-token distribution restricted to $\mathcal{V}_{\mathrm{num}}$ (renormalized on $\mathcal{V}_{\mathrm{num}}$), while the standard cross-entropy $\mathcal{L}_{\mathrm{CE}}$ is computed over the full vocabulary $\mathcal{V}$ using $\tilde{\mathbf{p}}$. For positions with $y \notin \mathcal{V}_{\mathrm{num}}$, our numeric-aware term is set to zero.

Standard cross-entropy only rewards exact matches and treats all incorrect numeric tokens equally. Our goal is to introduce a numeric-aware training loss that respects the metric structure of $\{v_i\}_{i=1}^N$: the loss should decrease as the predicted numeric distribution approaches the target distribution, while remaining compatible with token-level autoregressive training.

### 3.1. Distance-induced Kernel over Numeric Tokens

Numeric tokens come with an inherent geometry through their underlying values: for example, 3 is closer to 4 than to 9 in value space. We encode this geometry as a similarity kernel over $\mathcal{V}_{\mathrm{num}}$ by mapping pairwise value distances to kernel weights.

For indices $i, j \in \{1, \ldots, N\}$, define the value distance

$$d(i, j) = |v_i - v_j|. \tag{2}$$

A PSD kernel matrix is then obtained by converting distances into similarities with a (possibly multi-scale) radial kernel:

$$K_{ij} = \frac{1}{|\Sigma|} \sum_{\sigma \in \Sigma} \kappa_\sigma(d(i, j)), \tag{3}$$

where $\Sigma$ is a finite set of bandwidths and $\kappa_\sigma(\cdot)$ assigns higher similarity to numerically closer tokens. Throughout this paper, $\kappa_\sigma$ is instantiated as the Radial Basis Function (RBF),

$$\kappa_\sigma(d) = \exp\left(-\frac{d^2}{2\sigma^2}\right). \tag{4}$$

Intuitively, $\sigma$ controls the locality: smaller $\sigma$ makes similarity decay faster with $|v_i - v_j|$, while larger $\sigma$ couples farther-apart values. Since the RBF kernel is PSD on $\mathbb{R}$ and nonnegative averages preserve positive semidefiniteness, the resulting Gram matrix $\mathbf{K} \in \mathbb{R}^{N \times N}$ is symmetric PSD with $K_{ii} = 1$.

**Practical note on kernel K.** In modern LLMs, $\mathcal{V}_{\text{num}}$ typically consists of digit tokens (so $N = 10$). More generally, even when multi-digit integers are included as single tokens (e.g., $\{0, \ldots, 999\}$), $N$ is at most $10^3$. Thus the kernel can be precomputed once per tokenizer and reused throughout training with negligible overhead.

### 3.2. Kernel MMD Alignment

With the kernel-induced similarity structure over numeric tokens, the goal is to align the predicted numeric distribution $\mathbf{p}$ to the one-hot target $\mathbf{q}$ in a *value-aware* manner. Maximum Mean Discrepancy (MMD) (Gretton et al., 2012) provides a principled way to compare distributions through their kernel mean embeddings. Given a PSD kernel $k(\cdot, \cdot)$ with RKHS $\mathcal{H}$, the squared MMD between $P$ and $Q$ is

$$\text{MMD}^2(P, Q) = \|\mu_P - \mu_Q\|_{\mathcal{H}}^2, \tag{5}$$

where $\mu_P = \mathbb{E}_{x \sim P}[\phi(x)]$ and $\phi(\cdot)$ is the (implicit) feature map of $k$.

In our setting, the domain is the finite index set $\mathcal{X} = \{1, \ldots, N\}$, and both distributions are discrete vectors $\mathbf{p}, \mathbf{q} \in \Delta^N$ defined in (1). Let $\mathbf{K} \in \mathbb{R}^{N \times N}$ denote the Gram matrix, $K_{ij} = k(i, j)$. Introduce the prediction–target residual

$$\mathbf{r} = \mathbf{p} - \mathbf{q} \in \mathbb{R}^N. \tag{6}$$

Instantiating Eq. (5) on the finite domain yields a quadratic alignment loss in this residual:

$$\mathcal{L}_{\text{MMD}} := \sum_{i=1}^{N} \sum_{j=1}^{N} K_{ij} \, r_i r_j = \mathbf{r}^\top \mathbf{K} \mathbf{r}. \tag{7}$$

A detailed derivation from the expectation form Eq. (5) to the discrete residual form Eq. (7) is deferred to Appendix B.1. Moreover, when $\mathbf{K}$ is positive definite, $\mathcal{L}_{\text{MMD}} = 0$ holds if and only if $\mathbf{p} = \mathbf{q}$, so the auxiliary term preserves the supervised optimum.

### 3.3. Smooth Regularization via Dirichlet Energy

While $\mathcal{L}_{\text{MMD}}$ aligns $\mathbf{p}$ to the one-hot target under the kernel-induced similarity structure, it does not explicitly enforce *local consistency* of the prediction error along nearby numeric values. In particular, the residual can still exhibit sharp, locally oscillatory patterns over the numeric vocabulary. To address this, we introduce an additional smoothness regularizer by viewing the numeric sub-vocabulary as a weighted graph whose edge weights are given by kernel $\mathbf{K}$.

Specifically, we penalize variations of the residual $\mathbf{r}$ across strongly connected nodes using the Dirichlet energy:

$$\mathcal{L}_{\text{smooth}} := \frac{1}{2} \sum_{i=1}^{N} \sum_{j=1}^{N} K_{ij} \big(r_i - r_j\big)^2. \tag{8}$$

This objective has an intuitive interpretation. When two numeric values are close (large $K_{ij}$), the penalty strongly discourages $r_i$ and $r_j$ from disagreeing, thereby promoting a locally coherent error profile over the number line; when they are far apart (small $K_{ij}$), the coupling is weak and the regularizer imposes little constraint.

Let $\deg_i = \sum_{j=1}^{N} K_{ij}$, define $\mathbf{D} = \text{diag}(\deg_1, \ldots, \deg_N)$, and the graph Laplacian $\mathbf{L} = \mathbf{D} - \mathbf{K}$. Then the Dirichlet energy then reduces to the Laplacian quadratic form

$$\mathcal{L}_{\text{smooth}} = \mathbf{r}^\top \mathbf{L} \mathbf{r}, \tag{9}$$

highlighting that the regularizer suppresses high-frequency components of $\mathbf{r}$ on the kernel graph. The equivalence between the Dirichlet energy and the Laplacian quadratic form is a well-established identity, detailed in Appendix B.2 for completeness.

**Rationale for smoothing the residual.** An important design choice is to apply smoothness to the residual $\mathbf{r} = \mathbf{p} - \mathbf{q}$ rather than to $\mathbf{p}$ itself. Doing so preserves consistency with supervision: at perfect prediction $\mathbf{p} = \mathbf{q}$, the residual vanishes and $\mathcal{L}_{\text{smooth}} = 0$ automatically. By contrast, smoothing $\mathbf{p}$ directly would generally impose a nonzero penalty even when the model matches the target exactly, preventing the auxiliary objective from decaying to zero as training converges.

### 3.4. Unified Training Objective

We combine kernel MMD alignment and smoothness regularization into a single numeric-aware objective:

$$\begin{aligned}
\mathcal{L}_{\text{SMMD}} &= \mathbf{r}^\top \mathbf{K} \mathbf{r} + \alpha \, \mathbf{r}^\top \mathbf{L} \mathbf{r} \\
&= \mathbf{r}^\top \big(\mathbf{K} + \alpha \mathbf{L}\big) \mathbf{r},
\end{aligned} \tag{10}$$

where $\alpha$ controls the smoothness regularization strength. In practice, $\alpha$ is *automatically* set by a degree-based normalization, $\alpha = 1/(2\bar{d})$ with $\bar{d} = \frac{1}{N} \sum_{i=1}^{N} \deg_i$.

At non-numeric target positions, i.e. $y \notin \mathcal{V}_{\text{num}}$, $\mathcal{L}_{\text{SMMD}}$ is always set to 0. The overall training objective augments the standard cross-entropy with our SMMD term:

$$\mathcal{L} = \mathcal{L}_{\text{CE}} + \lambda \, \mathcal{L}_{\text{SMMD}}, \tag{11}$$

where $\lambda \geq 0$ controls the weight of the numeric-aware term. Pseudo-code for training with SMMD is provided in Appendix A. Both components are differentiable with respect to logits through $\mathbf{p} = \text{softmax}(\cdot)$ on numeric-target positions.

## 4. Experiments

This section presents a comprehensive evaluation of SMMD. Our findings show that:

1. SMMD consistently improves numerical prediction across language-only and vision-language tasks, outperforming both cross-entropy and recent numeric-target objectives.

2. Ablations verify that gains come from the value-distance construction of the kernel and the smooth residual regularizer, rather than from adding an MMD-style penalty alone.

3. Sensitivity studies show SMMD is robust under simple default settings, while the best $\lambda$ and $\sigma$ can vary across datasets.

### 4.1. Setup

**Tasks and datasets.** We evaluate SMMD on four numerical-output task categories spanning both language-only and vision-language settings. For *Mathematical Reasoning*, we train and evaluate on GSM8K (Cobbe et al., 2021), and further assess cross-dataset generalization on SVAMP (Patel et al., 2021). For *Arithmetic Calculation*, we use the DeepMind-Math suite (Saxton et al., 2019) (the arithmetic_mixed subset), which consists of short mixed-operator expressions (addition, subtraction, multiplication, and division) with numeric answers. To test grounded numerical prediction with visual inputs, we consider *Clock-Time Recognition* on the Clock-Time dataset (gpiosenka, 2022), where models predict the corresponding time given a clock image. We also evaluate *Chart Question Answering* on ChartQA (Masry et al., 2022), which requires extracting numeric values from plots and tables to answer questions. Across all datasets, we report exact-match accuracy for numeric outputs (with additional task-specific metrics reported where applicable). Additional dataset statistics and details are deferred to Appendix C.1.

**Models.** To assess the robustness and model-agnostic nature of SMMD across scales and architectures, we experiment with open-weight backbones ranging from 0.5B to 11B parameters. For LLMs, we consider Qwen2.5-0.5B and Qwen2.5-1.5B (Team, 2025), SmolLM3-3B (Bakouch et al., 2025), and Llama3-8B (Grattafiori et al., 2024). For VLMs, we choose Qwen2.5-VL-3B and Qwen2.5-VL-7B (Bai et al., 2025), Ministral-3-3B (Liu et al., 2026), and Llama-3.2-11B-Vision (Meta, 2024). [1] For ablation and analysis, we use representative backbones to keep the study focused and comparable: Qwen2.5-1.5B for LLM-based tasks and Qwen2.5-VL-3B for VLM-based tasks.

---

[1] Tokenizer note. Qwen2.5 series and Ministral-3 use digit-level tokenization for integers, so $\mathcal{V}_{\text{num}} = \{0, \ldots, 9\}$ and $N = 10$. SmolLM3 and Llama3 series include multi-digit integer tokens (e.g., $\{0, \ldots, 999\}$) as single tokens, yielding $N = 1000$ under our numeric-token construction.

### 4.2. Baselines

Beyond standard cross-entropy (CE), we compare our method against three recent numeric-target methods. The implementation details and hyperparameter settings are provided in Appendix C.2.

**Gaussian Cross Entropy (GCE)** (Wang et al., 2025) replaces the one-hot target for numeric tokens with a Gaussian-shaped soft target centered at the ground-truth numeric value, so that nearby numbers receive partial credit. Using the same notation as our method, let $y \in \mathcal{V}_{\text{num}}$ be the ground-truth numeric token with index $\pi(y)$ and value $v_{\pi(y)}$. GCE defines a soft target $\mathbf{q} \in \Delta^N$ over $\mathcal{V}_{\text{num}}$ by

$$q_i \propto \exp\Big( - \frac{(v_i - v_{\pi(y)})^2}{2\sigma_{\text{gce}}^2} \Big), \quad \sum_{i=1}^{N} q_i = 1, \quad (12)$$

and applies cross-entropy between $\mathbf{q}$ and the model distribution restricted to $\mathcal{V}_{\text{num}}$. Following Wang et al. (2025), we set $\sigma_{\text{gce}} = 0.5$ for GCE in all experiments.

**Number Token Loss (NTL)** (Zausinger et al., 2025) introduces a regression-like loss on numeric tokens by explicitly penalizing numeric distance. Following the paper, we use the Wasserstein-1 variant as our NTL baseline. In discrete numeric vocabulary with a one-hot target, the Wasserstein-1 objective reduces to a distance-weighted penalty that sums the predicted probability mass at each numeric token weighted by its absolute distance to the ground-truth value.

**Numeric Token Integrity Loss (NTIL)** (Fei et al., 2025) extends distance-aware supervision to *sequential* numeric prediction. Beyond token-level objectives, it adds (i) position-dependent weights reflecting place-value importance, and (ii) sequence-level consistency terms that use Gumbel-Softmax to reconstruct a differentiable scalar number and penalize magnitude/absolute errors of the reconstructed value. However, due to the requirement for dynamic sequence parsing and the non-vectorized nature of its number reconstruction process, NTIL incurs a computational latency several times higher than that of standard Cross-Entropy or plain EMD.

### 4.3. Main Results

Unless otherwise specified, we use a single RBF kernel with bandwidth $\sigma = 2.0$ and set our loss weight to $\lambda = 3.0$ across all experiments in this section. Additional implementation details are provided in Appendix C.3.

**Mathematical Reasoning.** Table 1 reports exact-match accuracy on GSM8K, and also evaluates the same models on SVAMP (full set) as a cross-dataset check. Across backbones, SMMD improves over CE on both datasets, and often outperforms prior numeric-target objectives. On

*Table 1.* **Mathematical reasoning results.** Metric: exact-match accuracy (Acc, %) ↑. We fine-tune on GSM8k training set, then evaluate on GSM8K test set and SVAMP full set.

| Model | Params | GSM8K | | | | | SVAMP | | | | |
|-------|--------|------|------|------|------|------|------|------|------|------|------|
| | | CE | GCE | NTL | NTIL | Ours | CE | GCE | NTL | NTIL | Ours |
| Qwen2.5 | 0.5B | 29.34 | 28.20 | 30.25 | 30.78 | **31.31** | 39.80 | 37.90 | 43.20 | 40.60 | **43.90** |
| Qwen2.5 | 1.5B | 54.99 | 52.69 | 56.17 | 55.19 | **57.77** | 68.60 | **71.10** | 59.80 | 67.10 | **71.10** |
| SmolLM3 | 3B | 67.78 | 66.56 | 64.13 | 70.13 | **70.74** | 65.80 | 63.70 | 60.50 | 65.80 | **67.10** |
| Llama3 | 8B | 55.72 | **58.15** | 48.12 | 56.63 | 57.70 | 62.20 | 58.40 | 57.50 | 59.50 | **65.30** |

GSM8K, the gains are consistent: for example, Qwen2.5-1.5B increases from 54.99% to 57.77% Acc, surpassing both NTL (56.17%) and NTIL (55.19%). Importantly, the gains transfer to SVAMP without additional training, indicating improved cross-dataset numerical generalization. For instance, Llama3-8B rises from 62.20% to 65.30% Acc. Overall, these results suggest that numeric-aware supervision translates into end-to-end reasoning gains and better robustness across datasets, rather than only reshaping local token probabilities. Appendix C.7 provides a statistical error analysis on GSM8K, which further quantifies how SMMD reduces common numerical error patterns.

**Arithmetic Calculation.** We evaluate direct arithmetic computation on the DeepMind-Math dataset. Exact-match accuracy (Acc, %) is the primary metric, and we additionally report mean absolute error (MAE) and the coefficient of determination ($R^2$) to capture both the magnitude and overall consistency of numerical deviations. As shown in Table 2, SMMD achieves the highest Acc among all compared methods across all four backbones. Moreover, for most models it also yields lower MAE and higher $R^2$, indicating that SMMD not only increases the likelihood of producing the exact answer, but also reduces large-magnitude mistakes when predictions are incorrect.

**Clock Time Recognition.** On the Clock-Time dataset, we evaluate grounded time prediction from clock images. We report exact-match accuracy (Acc, %) and *Time Gap*, defined as the absolute deviation in minutes. Table 3 shows that SMMD delivers strong and consistent improvements across VLM backbones: it attains the best Acc on three out of four models and remains competitive on the remaining one, suggesting the gains are not architecture-specific. Notably, Acc increases together with a clear reduction in Time Gap, indicating that SMMD reduces large time mistakes rather than merely shifting borderline cases. For example, on Ministral-3-3B it improves Acc from 82.36% to 97.56% and cuts Time Gap from 37.36 to 8.07 minutes. On Llama-3.2-Vision, SMMD remains highly competitive, trailing the strongest baseline by only 1.39 percentage points. A plausible reason is that this backbone is already well-calibrated on numeric tokens for this task, so GCE's confidence shaping fits its error profile and leaves less headroom

*Table 2.* **Arithmetic calculation results.** Metrics: exact-match accuracy (Acc, %) ↑, mean absolute error (MAE) ↓, and $R^2$ ↑.

| Model | Params | Loss | Acc | MAE | $R^2$ |
|-------|--------|------|-----|-----|-------|
| Qwen2.5 | 0.5B | CE | 43.39 | 3.38 | 0.67 |
| | | GCE | 42.04 | 3.32 | **0.73** |
| | | NTL | 44.79 | 3.56 | 0.67 |
| | | NTIL | 43.03 | 3.76 | 0.64 |
| | | Ours | **45.93** | **3.25** | 0.69 |
| Qwen2.5 | 1.5B | CE | 51.83 | 2.62 | 0.76 |
| | | GCE | 52.36 | **2.54** | **0.78** |
| | | NTL | 52.05 | 2.92 | 0.71 |
| | | NTIL | 52.99 | 2.60 | 0.75 |
| | | Ours | **53.83** | 2.66 | 0.77 |
| SmolLM3 | 3B | CE | 61.94 | 2.07 | 0.80 |
| | | GCE | 60.45 | 2.21 | 0.80 |
| | | NTL | 60.02 | 2.36 | 0.79 |
| | | NTIL | 61.28 | 2.14 | 0.81 |
| | | Ours | **64.06** | **1.88** | **0.82** |
| Llama3 | 8B | CE | 70.04 | 1.28 | 0.91 |
| | | GCE | 69.59 | 1.19 | 0.91 |
| | | NTL | 67.01 | 1.66 | 0.86 |
| | | NTIL | 67.91 | 1.66 | 0.85 |
| | | Ours | **71.96** | **1.16** | **0.92** |

for SMMD. These improvements are further corroborated by a digit-distribution probe: Figure 2 visualizes predicted digit probabilities under a fixed ground-truth digit condition. Compared to CE and other baselines, SMMD yields the sharpest and most target-aligned distribution, placing dominant probability mass on the correct digit while aggressively down-weighting nearby alternatives. This indicates a more decisive and stable numerical belief state—the model commits to the target with higher confidence and exhibits reduced ambiguity among competing digits. The complete per-bucket distributions for all ground-truth digits are deferred to Appendix C.6.

**Chart Question Answering.** We evaluate grounded numerical prediction from plots and tables on ChartQA dataset. Table 4 shows that all objectives are broadly comparable on this benchmark, suggesting that performance is often bottlenecked by visual grounding and value extraction rather than the numeric training signal alone. Nevertheless, SMMD remains consistently competitive and achieves the best or

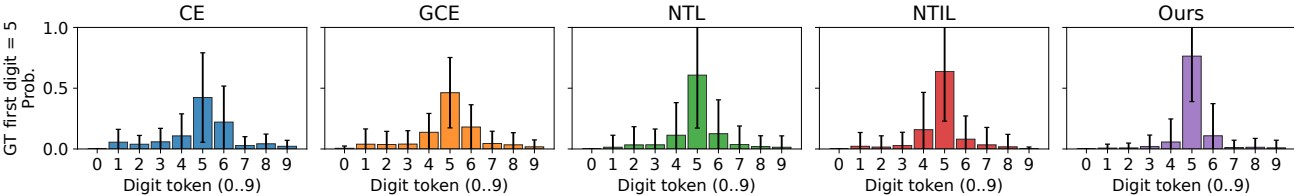

*Figure 2.* **Clock-Time digit distribution.** We evaluate model confidence by grouping test examples where the ground-truth first digit is 5. Using Qwen2.5-VL-3B as a representative VLM, we plot the averaged per-digit probability at the first-digit position. Compared to baselines, SMMD yields the most concentrated and target-aligned distribution, successfully assigning dominant probability mass to the correct digit while suppressing competing ones.

*Table 3.* **Clock-Time results.** Metrics: exact-match accuracy (Acc, %) ↑ and Time Gap ↓ (absolute deviation in minutes).

| Model | Params | Loss | Acc | Time Gap |
|---|---|---|---|---|
| | | CE | 46.39 | 86.73 |
| | | GCE | 38.40 | 55.28 |
| Qwen2.5-VL | 3B | NTL | 68.68 | 57.73 |
| | | NTIL | 69.46 | 55.43 |
| | | Ours | **74.30** | **50.59** |
| | | CE | 82.36 | 37.36 |
| | | GCE | 93.82 | 10.39 |
| Ministral-3 | 3B | NTL | 89.17 | 25.48 |
| | | NTIL | 90.21 | 23.88 |
| | | Ours | **97.56** | **8.07** |
| | | CE | 65.69 | 59.11 |
| | | GCE | 69.23 | 42.95 |
| Qwen2.5-VL | 7B | NTL | 75.97 | 41.45 |
| | | NTIL | 75.28 | 45.74 |
| | | Ours | **78.26** | **41.07** |
| | | CE | 71.80 | 46.64 |
| | | GCE | **82.15** | **28.63** |
| Llama-3.2-Vision | 11B | NTL | 79.79 | 33.29 |
| | | NTIL | 69.51 | 55.09 |
| | | Ours | 80.76 | 32.45 |

*Table 4.* **Chart question answering results.** Metric: exact-match accuracy (Acc, %) ↑.

| Model | CE | GCE | NTL | NTIL | Ours |
|---|---|---|---|---|---|
| Qwen2.5-VL (3B) | 71.44 | 70.44 | 72.01 | **72.95** | 72.21 |
| Ministral-3 (3B) | 71.64 | 72.32 | 69.66 | 71.95 | **73.21** |
| Qwen2.5-VL (7B) | 77.02 | 75.87 | 76.92 | 77.08 | **78.38** |
| Llama-3.2-Vision (11B) | 64.86 | **64.91** | 63.50 | 64.70 | **64.91** |

**MMD and smoothness contribute in complementary regimes.** Table 5 ablates SMMD by enabling the MMD term and the smoothness regularizer individually or jointly (CE disables both). Overall, the full objective performs best: it achieves the top accuracy on GSM8K, SVAMP, and ChartQA, and is effectively tied on Arithmetic. The MMD term contributes most on language-only reasoning (GSM8K/SVAMP), while smoothness is especially helpful on Arithmetic and Clock-Time, where local numeric consistency matters. Clock-Time is the only notable exception where adding MMD on top of smoothness slightly lowers Acc. This likely reflects that the task favors very sharp bucket decisions, and the additional MMD coupling can mildly blur near-target alternatives once smoothness has already stabilized the residual. Taken together, these results suggest that MMD and smoothness play complementary roles, with their combination offering the most reliable gains across diverse numerical prediction settings.

**Many kernels help somewhat, but distance-induced kernel helps the most.** We ask whether the gains come from the *distance-induced structure* encoded in the kernel $\mathbf{K}$, rather than from introducing an MMD-style loss alone. To isolate kernel design, we run all variants with the *MMD-only* objective (Eq. (7)), i.e., we drop the smoothness term so performance differences can be attributed to how $\mathbf{K}$ is constructed. We compare three settings: (i) **Distance-induced (Ours)**, where $K_{ij} = k(|v_i - v_j|)$ is built from true numerical distances; (ii) **Random PSD kernel**, which preserves the symmetric PSD structure but removes any link to nu-

tied-best accuracy on three of the four backbones, including a clear gain on Qwen2.5-VL-7B from 77.02% to 78.38%. While the absolute improvements are modest, the trend is stable: SMMD does not degrade ChartQA performance and can provide small, reliable gains in grounded settings.

### 4.4. Ablation Study and Analysis

This section presents ablations and analyses to understand *how* SMMD improves numerical prediction. We focus on targeted controlled studies that isolate key design choices: we (i) ablate the MMD term and the smoothness regularizer to quantify their respective contributions, (ii) vary the kernel construction to test whether value-induced distance structure is essential beyond the MMD form itself, and (iii) examine robustness to the main hyperparameters ($\sigma$ and $\lambda$). We summarize the main takeaways below.

*Table 5.* **Effect of MMD and smoothness terms.** We report performance when enabling the MMD term and the smoothness regularizer individually or jointly (SMMD), keeping all other settings fixed. Metric: exact-match accuracy (Acc, %) ↑.

| MMD | Smooth Reg | GSM8K | SVAMP | Arithmetic | Clock-Time | ChartQA |
|---|---|---|---|---|---|---|
| ✗ | ✗ | 54.97 | 68.60 | 51.83 | 46.39 | 71.44 |
| ✓ | ✗ | 57.16 | 70.10 | 52.37 | 68.61 | 71.95 |
| ✗ | ✓ | 56.48 | 68.70 | **53.84** | **75.14** | 71.95 |
| ✓ | ✓ | **57.77** | **71.10** | 53.83 | 74.30 | **72.21** |

*Table 6.* **Kernel-structure ablation (MMD-only).** We vary the kernel **K** while optimizing only the MMD term in Eq. (7) (smoothness removed), so differences isolate the effect of distance-induced kernel construction. Metrics are exact-match accuracy (Acc, %) ↑.

| Kernel setting | GSM8K | SVAMP | Arithmetic | Clock-Time | ChartQA |
|---|---|---|---|---|---|
| Random PSD kernel | 55.65 | 69.90 | 51.76 | 62.01 | 71.64 |
| Shuffled mapping | 56.17 | 69.90 | 51.98 | 67.98 | 71.74 |
| Distance-induced (Ours) | **57.16** | **70.10** | **52.37** | **68.61** | **71.95** |

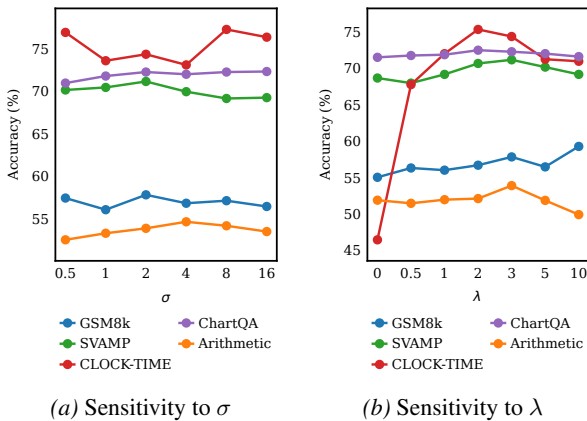

*(a)* Sensitivity to $\sigma$      *(b)* Sensitivity to $\lambda$

*Figure 3.* **Sensitivity analysis.** (a) Sensitivity to $\sigma$ (single-kernel). Accuracy versus the Gaussian bandwidth $\sigma$ in Eq. (3) with $\Sigma = \{\sigma\}$. Performance is typically highest near $\sigma=2.0$; overly small or large bandwidths reduce effectiveness by making the kernel too local or too diffuse. (b) Sensitivity to $\lambda$. Accuracy as we vary the SMMD weight $\lambda$ in Eq. (11). SMMD improves over $\lambda=0$ across a broad range and typically peaks near $\lambda=3$. GSM8K benefits from a larger $\lambda$ under its text-heavy reasoning trajectories.

meric values (detailed in Appendix C.4); and (**iii**) **Shuffled mapping**, which keeps the same distance-based functional form but permutes the value→token correspondence, breaking semantic alignment while preserving the overall kernel shape. As shown in Table 6, the distance-induced kernel is the only variant that is consistently best across all datasets, with particularly clear margins on Clock-Time. This pattern indicates that SMMD benefits from aligning distribution matching to *true numeric proximity*, rather than from a generic MMD regularization effect.

**A single well-chosen bandwidth is more important than multi-kernel mixtures.** We analyze the kernel bandwidth $\sigma$ in Eq. (3), which controls how rapidly similarity decays with value distance and thus how local the induced structure

is. Figure 3a reports results for the *single-kernel* setting ($\Sigma = \{\sigma\}$). Across datasets, performance is typically best around $\sigma=2.0$: smaller values make the kernel overly local (rewarding only near-identical numbers), while larger values make it overly diffuse (blurring meaningful proximity). Although the exact optimum can shift mildly by task, $\sigma=2.0$ is a strong and stable default that consistently improves over CE, so we use it throughout our main experiments. We additionally evaluated multi-kernel mixtures with bandwidths centered around 2.0 (Appendix C.5). However, these bring little benefit and can slightly underperform the best single-$\sigma$ choice, indicating that selecting a well-calibrated bandwidth matters more than increasing kernel complexity.

**SMMD is robust to $\lambda$ with a broad sweet spot.** We examine sensitivity to the SMMD weight $\lambda$ in the training objective in Eq. (11). Figure 3b shows that SMMD improves over the $\lambda=0$ baseline across tasks for a wide range of values, indicating that the method is not brittle to precise tuning. Performance typically peaks at a moderate $\lambda$: increasing $\lambda$ strengthens the numeric-target signal, while excessively large $\lambda$ can over-emphasize it and slightly hurt generalization. Across SVAMP and Clock-Time, the best performance is usually achieved around $\lambda=3$, and we use this value throughout our main experiments on all datasets. GSM8K is a slight exception, where gains continue to accrue at larger $\lambda$, consistent with its longer, text-heavy reasoning trajectories in which numeric targets form a smaller fraction of tokens.

**SMMD incurs only modest training overhead.** Finally, we measure the end-to-end training overhead of SMMD under a controlled GSM8K setup. Although SMMD involves quadratic forms $r^\top K r$ and $r^\top L r$ over the numeric sub-vocabulary, both $K$ and $L$ are precomputed once, and the numeric vocabulary is small in practice. As shown in Table 7, SMMD adds only a small wall-clock overhead

*Table 7.* End-to-end training overhead on GSM8K. We report average step time and peak allocated GPU memory on a single NVIDIA L20 GPU. For each loss, we run 120 training steps, discard the first 20 warm-up steps, and report mean step time over 3 repeated runs.

| Model | Loss | Step time (ms) ↓ | Mem. (GB) ↓ |
|---|---|---|---|
| Qwen2.5-1.5B | CE | $322.29 \pm 2.97$ | 8.02 |
| | NTL | $357.77 \pm 0.77$ | 10.27 |
| | NTIL | $929.07 \pm 3.19$ | 10.27 |
| | SMMD | $333.21 \pm 1.93$ | 10.28 |
| SmolLM3-3B | CE | $439.82 \pm 0.74$ | 7.05 |
| | NTL | $468.98 \pm 2.92$ | 8.70 |
| | NTIL | $924.49 \pm 0.84$ | 8.71 |
| | SMMD | $449.40 \pm 0.48$ | 8.71 |

over CE: 3.39% on Qwen2.5-1.5B with a digit-level tokenizer and 2.18% on SmolLM3-3B with a multi-digit tokenizer. This overhead is substantially lower than NTIL, whose sequence-level reconstruction incurs much higher latency. The memory increase is comparable to other numeric-aware losses, suggesting that SMMD remains practical for downstream fine-tuning.

## 5. Conclusion

We studied numerical prediction in LLMs through the lens of objective mismatch: although numbers exhibit an inherent metric structure, standard cross-entropy treats numeric tokens as unstructured categories. To bridge this gap, we introduced **SMMD**, a plug-and-play training loss that constructs a distance-induced kernel over a numeric sub-vocabulary, aligns the predicted numeric distribution with the target via kernel MMD, and regularizes the prediction–target residual on the induced kernel graph. Across diverse language-only and vision-language tasks with numeric targets, SMMD consistently improves numerical accuracy over cross-entropy and strong numeric-target baselines. Ablations further confirm our kernel design and smoothness term contribute complementary benefits.

Despite these gains, SMMD has limitations in two respects. First, its notion of distance is defined over token-level numeric units and is therefore constrained by the tokenizer; under digit-level tokenization, nearby integers can differ across multiple digits, and a per-token objective may over-penalize such near-miss cases. This distance-based formulation also assumes that numerical proximity is semantically meaningful for the supervised target, an assumption that may not hold when numbers primarily serve as identifiers or symbolic labels, such as category IDs or codes. Second, SMMD introduces additional hyperparameters: the loss weight $\lambda$ and the kernel bandwidth $\sigma$. While our defaults work well broadly, optimal settings can vary by task, motivating adaptive hyperparameter selection in future work.

## Acknowledgements

This work was supported by the National Key R&D Program of China (2024YFB3312503), the Science and Technology Major Project of Sichuan Province (2024ZDZX0003, 2025ZDZX0140). We also acknowledge the support of Sichuan Province Engineering Technology Research Center of Broadband Electronics Intelligent Manufacturing.

## Impact Statement

This work improves the numerical reliability of large language models by introducing a training objective for numeric tokens, which can benefit applications where accurate quantities matter such as education and data analysis. While the method reduces common numerical errors, it does not guarantee correctness in all cases and may increase the perceived authority of generated numbers. Therefore, outputs should be validated in high-stakes settings and deployed with standard safeguards such as human oversight and monitoring.

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

# A. Algorithms

---

**Algorithm 1** Constructing the numeric sub-vocabulary $\mathcal{V}_{\text{num}}$

---

**Require:** token vocabulary $\mathcal{V}$; deterministic numeric parser $\text{parse}(\cdot)$ (float casting)
**Ensure:** numeric sub-vocabulary $\mathcal{V}_{\text{num}}$; value map $\{v_i\}_{i=1}^N$; index map $\pi$

$\mathcal{V}_{\text{num}} \leftarrow \emptyset$
Initialize an empty map $\text{val}[\cdot]$
**for all** $t \in \mathcal{V}$ **do**
  **if** $\text{parse}(t)$ succeeds **then**
    $\text{val}[t] \leftarrow \text{parse}(t)$
    $\mathcal{V}_{\text{num}} \leftarrow \mathcal{V}_{\text{num}} \cup \{t\}$
  **end if**
**end for**
Choose any bijection $\pi : \mathcal{V}_{\text{num}} \rightarrow \{1, \dots, N\}$ with $N = |\mathcal{V}_{\text{num}}|$
**for** $i = 1$ **to** $N$ **do**
  $v_i \leftarrow \text{val}(\pi^{-1}(i))$
**end for**

---

**Algorithm 2** Training with SMMD

---

**Require:** training data $\mathcal{D}$; LM $p_\theta$; numeric sub-vocabulary $\mathcal{V}_{\text{num}}$ with index map $\pi$; precomputed $\mathbf{K}$, $\mathbf{L}$, and $\alpha$; weight $\lambda$
**Ensure:** trained parameters $\theta$

**while** not converged **do**
  Sample a minibatch and run a forward pass to obtain logits $\{\boldsymbol{\ell}_t\}$
  Compute $\mathcal{L}_{\text{CE}}$ over the full vocabulary $\mathcal{V}$
  Initialize $S \leftarrow 0, \quad M \leftarrow 0$         ($S$: loss sum, $M$: #numeric positions)
  **for all** token positions $t$ in the minibatch **do**
    **if** $y_t \in \mathcal{V}_{\text{num}}$ **then**
      $\mathbf{p}_t \leftarrow \text{softmax}(\boldsymbol{\ell}_t[\mathcal{V}_{\text{num}}]), \ \mathbf{r}_t \leftarrow \mathbf{p}_t - \mathbf{e}_{\pi(y_t)}$
      $S \leftarrow S + \mathbf{r}_t^\top \mathbf{K} \mathbf{r}_t + \alpha \, \mathbf{r}_t^\top \mathbf{L} \mathbf{r}_t$
      $M \leftarrow M + 1$
    **end if**
  **end for**
  $\mathcal{L}_{\text{SMMD}} \leftarrow \frac{S}{\max(1, M)}$
  Update $\theta$ by minimizing $\mathcal{L} \leftarrow \mathcal{L}_{\text{CE}} + \lambda \mathcal{L}_{\text{SMMD}}$
**end while**

---

# B. Mathematical Derivations of SMMD Objectives

This appendix provides detailed step-by-step derivations for the two components of our numeric-aware loss: the discrete MMD alignment (Eq. 14) and the smooth regularization via Dirichlet energy (Eq. 10).

## B.1. Derivation of Discrete MMD for Numeric Tokens

We show how the general MMD definition reduces to the quadratic form $\mathbf{r}^\top \mathbf{K} \mathbf{r}$ on a finite numeric vocabulary.

**1. General Expectation Form.** For two distributions $P$ and $Q$ in a RKHS $\mathcal{H}$ with kernel $k(\cdot, \cdot)$, the squared MMD is:

$$\text{MMD}^2(P, Q) = \mathbb{E}_{x, x' \sim P}[k(x, x')] - 2\mathbb{E}_{x \sim P, y \sim Q}[k(x, y)] + \mathbb{E}_{y, y' \sim Q}[k(y, y')]. \tag{13}$$

**2. Discrete Summation.** In our setting, the domain is the finite sub-vocabulary $\mathcal{V}_{\text{num}}$ of size $N$. The distributions are discrete vectors $\mathbf{p}, \mathbf{q} \in \Delta^N$. The expectations become double summations over indices $i$ and $j$:

$$\mathcal{L}_{\text{MMD}} = \sum_{i=1}^{N}\sum_{j=1}^{N} p_i p_j K_{ij} - 2\sum_{i=1}^{N}\sum_{j=1}^{N} p_i q_j K_{ij} + \sum_{i=1}^{N}\sum_{j=1}^{N} q_i q_j K_{ij}. \tag{14}$$

**3. Residual Vectorization.** Exploiting the linearity of summation and the symmetry of $\mathbf{K}$, we factorize the expression using the prediction–target residual $r_i = p_i - q_i$:

$$\mathcal{L}_{\text{MMD}} = \sum_{i=1}^{N}\sum_{j=1}^{N} K_{ij}(p_i p_j - p_i q_j - q_i p_j + q_i q_j) \tag{15}$$

$$= \sum_{i=1}^{N}\sum_{j=1}^{N} K_{ij}(p_i - q_i)(p_j - q_j) = \sum_{i=1}^{N}\sum_{j=1}^{N} K_{ij} r_i r_j. \tag{16}$$

This yields the compact quadratic form:

$$\mathcal{L}_{\text{MMD}} = \mathbf{r}^{\top}\mathbf{K}\mathbf{r}. \tag{17}$$

### B.2. Derivation of Graph Laplacian Form for Smooth Regularization

We now derive the matrix representation for the Dirichlet energy $\mathcal{L}_{\text{smooth}}$, which penalizes local variations of the residual $\mathbf{r}$ over the similarity graph $\mathbf{K}$.

**1. Expansion of Dirichlet Energy.** Starting from the pairwise difference penalty in Eq. (9):

$$\mathcal{L}_{\text{smooth}} = \frac{1}{2}\sum_{i=1}^{N}\sum_{j=1}^{N} K_{ij}(r_i - r_j)^2. \tag{18}$$

Expanding the quadratic term $(r_i - r_j)^2 = r_i^2 - 2r_i r_j + r_j^2$:

$$\mathcal{L}_{\text{smooth}} = \frac{1}{2}\left(\sum_{i,j} K_{ij} r_i^2 - 2\sum_{i,j} K_{ij} r_i r_j + \sum_{i,j} K_{ij} r_j^2\right). \tag{19}$$

**2. Relation to the Degree Matrix.** For the first term, summing over $j$ yields the degree of node $i$: $deg_i = \sum_{j=1}^{N} K_{ij}$. Thus:

$$\sum_{i=1}^{N}\sum_{j=1}^{N} K_{ij} r_i^2 = \sum_{i=1}^{N} r_i^2 \left(\sum_{j=1}^{N} K_{ij}\right) = \sum_{i=1}^{N} deg_i r_i^2 = \mathbf{r}^{\top}\mathbf{D}\mathbf{r}, \tag{20}$$

where $\mathbf{D} = \text{diag}(deg_1, \ldots, deg_N)$. Due to the symmetry of $\mathbf{K}$, the third term $\sum_{i,j} K_{ij} r_j^2$ also equals $\mathbf{r}^{\top}\mathbf{D}\mathbf{r}$.

**3. Final Laplacian Form.** Substituting these into the expression:

$$\mathcal{L}_{\text{smooth}} = \frac{1}{2}\left(\mathbf{r}^{\top}\mathbf{D}\mathbf{r} - 2\mathbf{r}^{\top}\mathbf{K}\mathbf{r} + \mathbf{r}^{\top}\mathbf{D}\mathbf{r}\right) \tag{21}$$

$$= \mathbf{r}^{\top}\mathbf{D}\mathbf{r} - \mathbf{r}^{\top}\mathbf{K}\mathbf{r} = \mathbf{r}^{\top}(\mathbf{D} - \mathbf{K})\mathbf{r}. \tag{22}$$

Defining the graph Laplacian $\mathbf{L} = \mathbf{D} - \mathbf{K}$, we obtain the final form:

$$\mathcal{L}_{\text{smooth}} = \mathbf{r}^{\top}\mathbf{L}\mathbf{r}. \tag{23}$$

Comparing the two objectives, while $\mathcal{L}_{\text{MMD}} = \mathbf{r}^{\top}\mathbf{K}\mathbf{r}$ captures global distribution alignment, $\mathcal{L}_{\text{smooth}} = \mathbf{r}^{\top}\mathbf{L}\mathbf{r}$ ensures that the residual is distributed smoothly across numerically similar tokens.

# C. Experiment

## C.1. Datasets

Figure 4 shows example question–answer pairs across datasets, and Table 8 summarizes the train/test sizes. While GSM8K and SVAMP provide intermediate solution rationales, we evaluate only the *final numeric answer* and compute exact-match accuracy based on this final output. Since our focus is numeric prediction, we preprocess ChartQA by filtering out examples whose ground-truth answers are non-numeric, such as yes/no judgments or free-form text. ChartQA contains 28.3k/2.5k train/test examples before filtering, and 22.8k/1.9k after filtering.

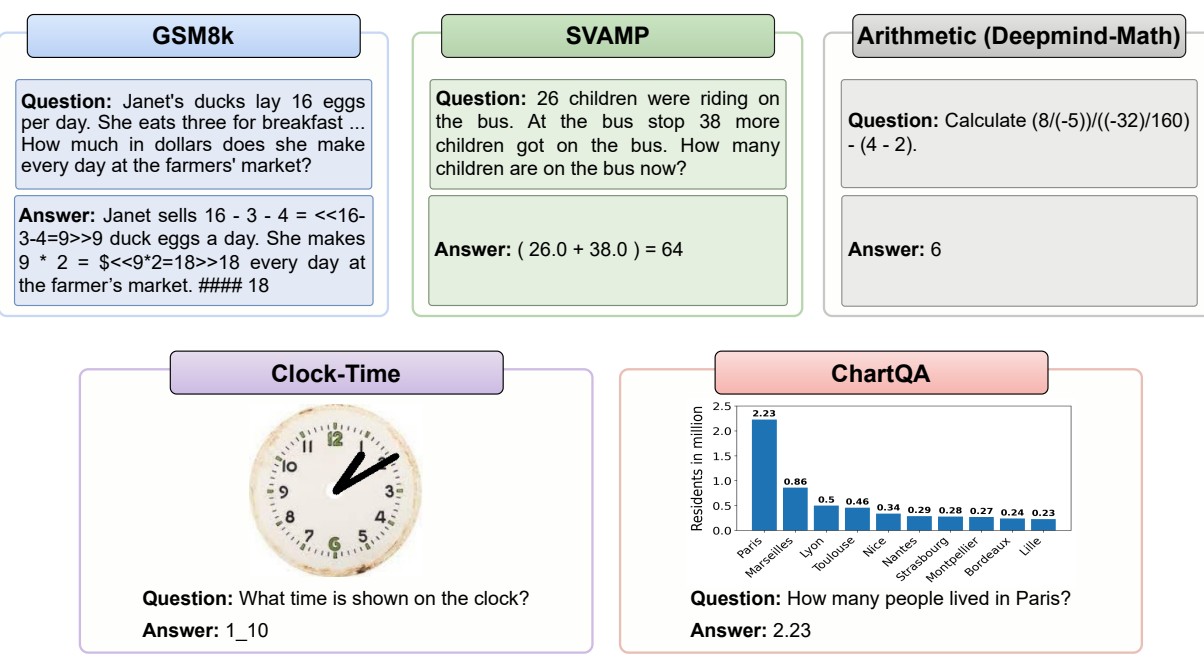

*Figure 4.* **Dataset Samples Illustration:** Question-Answer Pairs with Numeric Outputs

*Table 8.* **Dataset sizes used in our experiments.**

|              | GSM8k | SVAMP | Arithmetic | Clock-Time | ChartQA |
|--------------|-------|-------|------------|------------|---------|
| **Training set** | 7.47k | –     | 2M         | 11.52k     | 22.85k  |
| **Test set**     | 1.32k | 1k    | 10k        | 1.44k      | 1.92k   |

## C.2. Baselines

### C.2.1. NUMBER TOKEN LOSS (NTL)

We implement NTL using the Wasserstein-1 formulation in Zausinger et al. (2025) and add it to standard cross-entropy. Let $\mathcal{V}_{\text{num}} = \{u_i\}_{i=1}^N$ be the numeric-token subset, where each numeric token $u_i$ is mapped to a real value $v_i \in \mathbb{R}$ (via deterministic parsing of the token string). At a decoding step $t$ with ground-truth token $y_t \in \mathcal{V}_{\text{num}}$, we restrict the model distribution to $\mathcal{V}_{\text{num}}$ and denote it by $\mathbf{p} \in \Delta^N$.

NTL is defined as a Wasserstein-1 distance between the one-hot target and the predicted numeric-token distribution:

$$\mathcal{L}_{\text{NTL}} = \min_{\gamma \in \Gamma(\mathbf{e}_{\pi(y)}, \mathbf{p})} \sum_{i=1}^N \sum_{j=1}^N \gamma_{ij} \, |v_i - v_j|, \tag{24}$$

where $y \in \mathcal{V}_{\text{num}}$ is the ground-truth numeric token, $\pi(y)$ is its index in $\mathcal{V}_{\text{num}}$, $\mathbf{e}_{\pi(y)}$ is the one-hot target, and $\Gamma(\cdot, \cdot)$ is the set of couplings with the specified marginals. In our one-dimensional discrete setting with a one-hot target, Eq. (24) reduces

to a directly computable form:

$$\mathcal{L}_{\text{NTL}} = \sum_{i=1}^{N} p_i \, |v_i - v_{\pi(y)}|, \tag{25}$$

i.e., the expected absolute numeric deviation under $\mathbf{p}$, so probability mass assigned to numerically closer tokens incurs smaller penalty. We apply $\mathcal{L}_{\text{NTL}}$ only at positions where the ground-truth token is numeric (and set it to 0 otherwise), and combine it with cross-entropy:

$$\mathcal{L} = \mathcal{L}_{\text{CE}} + \lambda \, \mathcal{L}_{\text{NTL}}. \tag{26}$$

**Hyperparameters.** We perform a small, representative scan of the NTL loss weight $\lambda$ on two settings: Qwen2.5-1.5B on GSM8K and Qwen2.5-VL-3B on Clock-Time. We sweep a modest grid of $\lambda$ values (Figure 5) and observe that NTL reaches its best performance at $\lambda = 2.0$ on both datasets. Accordingly, we use $\lambda = 2.0$ as the default NTL setting in all experiments.

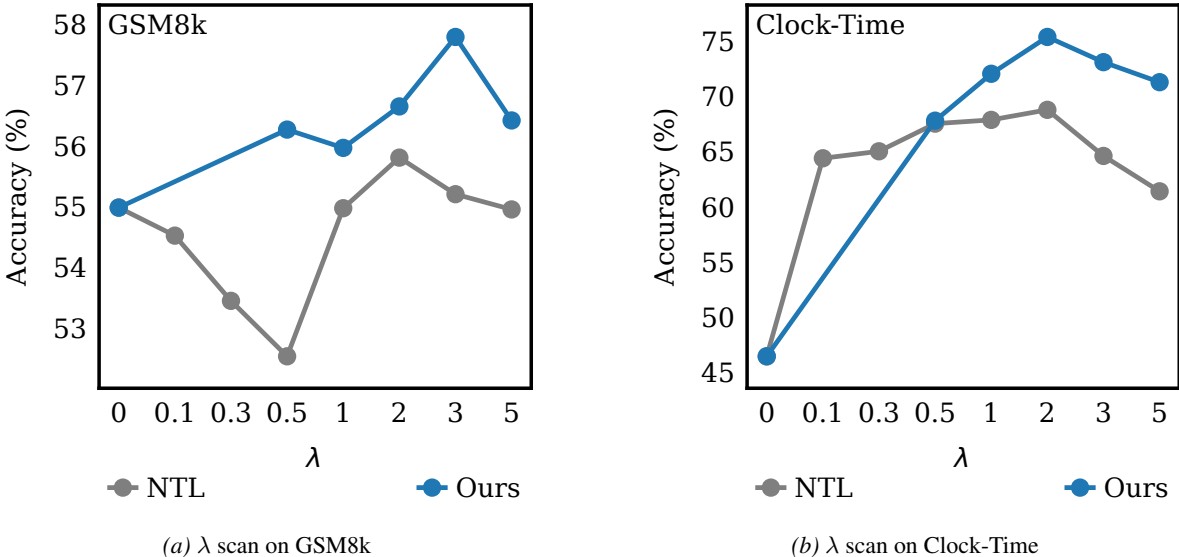

*(a)* $\lambda$ scan on GSM8k

*(b)* $\lambda$ scan on Clock-Time

*Figure 5.* $\lambda$ **scan for the NTL baseline (with SMMD (Ours) as reference).** We vary the NTL loss weight $\lambda$ and report exact-match accuracy (Acc, %) for (a) Qwen2.5-1.5B on GSM8K and (b) Qwen2.5-VL-3B on Clock-Time. NTL peaks at $\lambda = 2.0$ in both settings, which we adopt as the default for NTL throughout. The SMMD (Ours) curve is shown for reference under the same $\lambda$ values.

### C.2.2. NUMERIC INTEGRITY TOKEN LOSS (NTIL)

We implement NTIL following Fei et al. (2025). NTIL extends the token-level Wasserstein/EMD objective to *sequential* digit prediction by (i) emphasizing more significant digit positions and (ii) adding sequence-level penalties based on the *constructed* numeric value. Concretely, consider a ground-truth number span consisting of $n$ consecutive digit tokens with digit labels $\{d_k\}_{k=0}^{n-1}$, where $d_k \in \{0,\ldots,9\}$ and $k = 0$ is the most significant digit. Let $\mathbf{p}^{(k)} \in \Delta^{10}$ be the model distribution over digit tokens at position $k$. The token-level term is a 1D Wasserstein-1 / EMD with ground cost $c(i,j) = |i - j|$:

$$\mathcal{L}_{\text{EMD}} = \sum_{k=0}^{n-1} w_k \cdot \text{EMD}\big(\mathbf{e}_{d_k}, \mathbf{p}^{(k)}\big) = \sum_{k=0}^{n-1} w_k \sum_{i=0}^{9} p_i^{(k)} \, |i - d_k|, \qquad w_k = (1+\tau)^{n-k-1}, \tag{27}$$

where the equality uses the one-hot target simplification in the discrete 1D setting, and the exponential weights $w_k$ reflect place-value importance.

To enforce sequence-level numeric consistency, NTIL constructs a differentiable estimate of the *entire* predicted number via Gumbel-softmax over digits. Denote the resulting soft digit at position $k$ by $\hat{d}_k = \sum_{i=0}^{9} i \, \tilde{y}_i^{(k)}$, and aggregate across positions (using the corresponding powers of 10) to obtain a scalar prediction $X$; the ground-truth numeric value is $Y$. Two additional sequence-level losses are then applied:

$$\mathcal{L}_{\text{rel}} = \frac{|X - Y|}{\max(X,Y) + \epsilon}, \qquad \mathcal{L}_{\text{mag}} = \log\Big(\frac{\max(X,Y)}{\min(X,Y)}\Big), \tag{28}$$

with a small $\epsilon$ to avoid division by zero. The NTIL auxiliary objective is the sum of three terms:

$$\mathcal{L}_{\mathrm{NTIL}} = \mathcal{L}_{\mathrm{EMD}} + \alpha\,\mathcal{L}_{\mathrm{rel}} + \beta\,\mathcal{L}_{\mathrm{mag}}, \tag{29}$$

and the final training loss adds it to cross-entropy:

$$\mathcal{L} = \mathcal{L}_{\mathrm{CE}} + \lambda\,\mathcal{L}_{\mathrm{NTIL}}. \tag{30}$$

**Hyperparameters.** For the overall NTIL weight, we set $\lambda = 2.0$ in all experiments for consistency with our representative scan of EMD-based objectives (Figure 5); since NTIL's primary term is a weighted EMD loss (Eq. (27)), we adopt the same EMD weight under the same computational budget constraints. For the sequence-level terms, we follow the default settings in the original implementation and use $\alpha = \beta = \tau = 0.2$.

### C.3. Implementation Details

We fine-tune all backbones on NVIDIA A800 and L20 GPUs using the Unsloth framework (Daniel Han & team, 2023) and Transformers (Wolf et al., 2020) (v4.57.3; except Ministral-3, which requires Transformers v5.0.0+). All models are fine-tuned with LoRA (rank $r{=}16$) and optimized using AdamW with a learning rate of $2 \times 10^{-4}$ and weight decay 0.01; we use a linear warmup over the first 3% of training steps (warmup ratio 0.03). For the Arithmetic and Clock-Time datasets, we train for 15 epochs, as the training loss continues to decrease throughout training; for all other datasets, we train for 5 epochs. In every setting, we select the checkpoint with the best validation performance and report the corresponding test results. At inference time, we use greedy decoding for reproducibility.

### C.4. Construction of the random PSD kernel

Let $D$ be the size of the numeric sub-vocabulary and let $d$ be a small embedding dimension. We construct a random kernel matrix $\mathbf{K} \in \mathbb{R}^{D \times D}$ that is symmetric, positive semidefinite (PSD), and independent of the underlying numeric values. We sample i.i.d. random embeddings $\{z_i\}_{i=1}^{D}$ with $z_i \in \{-1, +1\}^d$ (Rademacher vectors), normalize each to unit $\ell_2$ norm, and stack them into a matrix $Z \in \mathbb{R}^{D \times d}$. In our experiments we use $d{=}4$, which yields a compact random feature space while ensuring $\|z_i\|_2 = 1$.

We first form the Gram matrix

$$\mathbf{K}^{(0)} = ZZ^{\top}, \tag{31}$$

which is PSD by construction and satisfies $K_{ii}^{(0)} = 1$. To increase the contrast of these random correlations while preserving PSD, we apply an elementwise (Hadamard) integer power:

$$\mathbf{K} = \left(\mathbf{K}^{(0)}\right)^{\circ p}. \tag{32}$$

Equivalently, each entry is given by $K_{ij} = (z_i^{\top} z_j)^p$. PSD is preserved because $\left(\mathbf{K}^{(0)}\right)^{\circ p}$ is the Hadamard product of $\mathbf{K}^{(0)}$ with itself $p$ times, and the Schur product theorem guarantees that Hadamard products of PSD matrices remain PSD. Throughout our ablations, we set the polynomial degree to the odd integer $p{=}5$, which strengthens these value-agnostic correlations while keeping the kernel PSD, providing a stringent control that removes number-line structure without changing the MMD form.

### C.5. Results of Multi-Kernel

Table 9 further examines whether multi-kernel mixtures provide additional benefits beyond a well-chosen single bandwidth. Starting from the best-performing single-kernel setting in the main paper ($\{\sigma\} = \{2.0\}$), we evaluate several representative bandwidth sets centered around 2.0 as well as wider and more diverse mixtures. Overall, the improvements from multi-kernel averaging are limited and inconsistent across tasks: the single bandwidth $\sigma = 2.0$ remains the best or second-best choice on most datasets, while mixtures sometimes yield marginal gains on a specific dataset (e.g., SVAMP) but often degrade performance elsewhere. These results suggest that, in our setting, accurately calibrating the kernel locality (via a single $\sigma$) is more important than increasing kernel complexity. Therefore, for simplicity and robustness, we adopt a single-kernel configuration with $\sigma = 2.0$ as the default throughout the paper.

*Table 9.* **Multi-kernel bandwidth study.** SMMD performance under different Gaussian bandwidth sets $\Sigma$ (single-kernel and multi-kernel mixtures) on the ablation backbones. Best is in **bold** and second-best is underlined.

| $\{\sigma\}$ | Qwen2.5-1.5B | | | Qwen2.5-VL-3B | |
|---|---|---|---|---|---|
| | GSM8K | SVAMP | Arithmetic | Clock-Time | ChartQA |
| $\{2.0\}$ | **57.77** | 71.10 | **53.83** | 74.30 | **72.22** |
| $\{1.7, 2.0, 2.3\}$ | 57.31 | **71.20** | 52.74 | 73.26 | 71.90 |
| $\{1.0, 2.0, 3.0\}$ | 57.16 | 70.70 | **53.83** | 72.29 | 72.06 |
| $\{1.0, 1.5, 2.0, 2.5, 3.0\}$ | 56.78 | 70.90 | 52.25 | 73.47 | 71.59 |
| $\{0.5, 3.0, 5.5\}$ | 56.33 | 70.60 | 51.98 | **74.79** | 71.33 |

## C.6. Numeric distribution analysis

Take Qwen2.5-VL-3B as example, we examine how different objectives shape the model's digit distributions on Clock-Time Dataset. We group examples by the first digit of the ground-truth hour and summarize the average digit distribution (mean $\pm$ std over examples) for each group. As shown in Figure 6, our method consistently places more probability on the correct target digit while suppressing competing digits, producing a sharper and more target-aligned distribution across most buckets.

## C.7. Error Analysis

We conduct a targeted error analysis on GSM8K to understand how our numeric-aware objective changes failure modes beyond exact-match accuracy. For each example, we parse the final numeric answer from the model output (and the ground-truth answer) and compute per-instance errors. All statistics below are reported on the *common-valid* subset where both methods yield a valid parsed number (1319 examples).

**Overall error distribution.** Figure 7 shows the histogram of $\log_{10}(|\hat{y} - y| + 1)$. Our method shifts the error mass toward smaller values and reduces the tail: the 90th percentile absolute error decreases from 168.40 (CE) to 150.00 (Ours), indicating fewer large-magnitude failures.

**Digit-sliced signed error histograms.** To probe digit-boundary behaviors, we slice GSM8K examples by the last digit of the ground-truth answer (0–9) and plot signed errors $(\hat{y} - y)$. Figure 8 shows that for most ending digits, our error mass is more concentrated near zero, aligning with higher exact/near-exact outcomes. We also observe a mild limitation on boundary-like endings (notably 0 and 9), where the two methods are comparable and occasionally CE is slightly better, suggesting that our gains mainly come from reducing off-scale errors rather than uniformly improving digit-boundary exactness.

**Scale-related failure modes.** A frequent failure in GSM8K is *off-scale* prediction (e.g., $\times 10$ due to unit/decimal mistakes) or *catastrophic* relative error. Table 10 categorizes each prediction on the GSM8K test set into mutually exclusive types: (i) `exact`: $\hat{y} = y$; (ii) `sign_flip`: $\hat{y} \approx -y$; (iii) `scale_x10^k`: $|\hat{y}| \approx |y| \cdot 10^k$ for $k \in \{-3, -2, -1, 1, 2, 3\}$ (within 5% relative tolerance); (iv) `near_miss`: $|\hat{y} - y| \leq 1$ or relative error $\leq 1\%$; (v) `catastrophic`: relative error $\geq 50\%$; (vi) `other`: remaining cases. Compared to CE, our method increases exact matches (726→762) and near-misses (32→41), while reducing catastrophic errors (331→283, a $\sim$14.5% relative reduction) and scale errors (16→10), especially $\times 10$ mistakes (9→4). This supports our main claim: metric-aligned supervision primarily suppresses off-scale guesses and shifts errors toward numerically plausible neighborhoods.

## C.8. SMMD is compatible with standard text modeling.

To sanity-check that SMMD does not interfere with general language generation, we first evaluate on SAMSum (Gliwa et al., 2019), a dialogue summarization benchmark scored by ROUGE. Since SMMD is only activated at positions whose targets fall in $\mathcal{V}_{\text{num}}$, it introduces no additional supervision on the vast majority of purely textual tokens in this task. As shown in Table 11, SMMD yields ROUGE scores that are very close to the CE baseline, with differences within 0.6 absolute across ROUGE variants.

We further evaluate whether the numeric-aware fine-tuning affects broader knowledge and reasoning ability. Using the same

*Table 10.* **Scale-error category breakdown on GSM8K** ($n = 1319$)**.** Our method reduces catastrophic relative errors and $\times 10^k$ scale mistakes (notably $\times 10$), while increasing exact and near-miss outcomes.

| Category | CE | | Ours | |
|---|---|---|---|---|
| | **#** | **%** | **#** | **%** |
| Exact | 726 | 55.04 | 762 | 57.77 |
| Sign flip | 0 | 0.00 | 2 | 0.15 |
| Scale $\times 10^k$ (any $k \neq 0$) | 16 | 1.21 | 10 | 0.76 |
| $\quad k = -2$ | 2 | 0.15 | 2 | 0.15 |
| $\quad k = -1$ | 3 | 0.23 | 3 | 0.23 |
| $\quad k = 1$ | 9 | 0.68 | 4 | 0.30 |
| $\quad k = 2$ | 1 | 0.08 | 1 | 0.08 |
| $\quad k = 3$ | 1 | 0.08 | 0 | 0.00 |
| Near-miss | 32 | 2.43 | 41 | 3.11 |
| Catastrophic | 331 | 25.09 | 283 | 21.46 |
| Other | 214 | 16.22 | 221 | 16.76 |
| Total | | | | 1319 |

*Table 11.* **SAMSum dialogue summarization.** Backbone: Qwen2.5-1.5B. Metrics: ROUGE $\uparrow$.

| Method | ROUGE-1 | ROUGE-2 | ROUGE-L |
|---|---|---|---|
| CE | 51.90 | 27.33 | 43.49 |
| Ours | 51.36 | 26.79 | 43.01 |

*Table 12.* **MMLU evaluation.** Backbone: Qwen2.5-1.5B fine-tuned on GSM8K. We report 5-shot accuracy (%) using `lm-eval`.

| Method | STEM | Humanities | Social Sci. | Other | Avg. |
|---|---|---|---|---|---|
| CE | $55.47 \pm 0.85$ | $54.94 \pm 0.68$ | $71.60 \pm 0.80$ | $66.72 \pm 0.82$ | $61.32 \pm 0.39$ |
| Ours | $55.44 \pm 0.86$ | $54.86 \pm 0.68$ | $71.79 \pm 0.80$ | $66.69 \pm 0.82$ | $61.32 \pm 0.39$ |

Qwen2.5-1.5B checkpoints fine-tuned on GSM8K, we evaluate MMLU under the standard 5-shot protocol with `lm-eval` (Gao et al., 2024). Table 12 shows that SMMD matches CE in average accuracy (61.32% for both methods), with nearly identical performance across all four category groups. Together, these results suggest that SMMD is broadly compatible with standard text modeling and does not materially degrade non-numeric capabilities in these sanity checks.

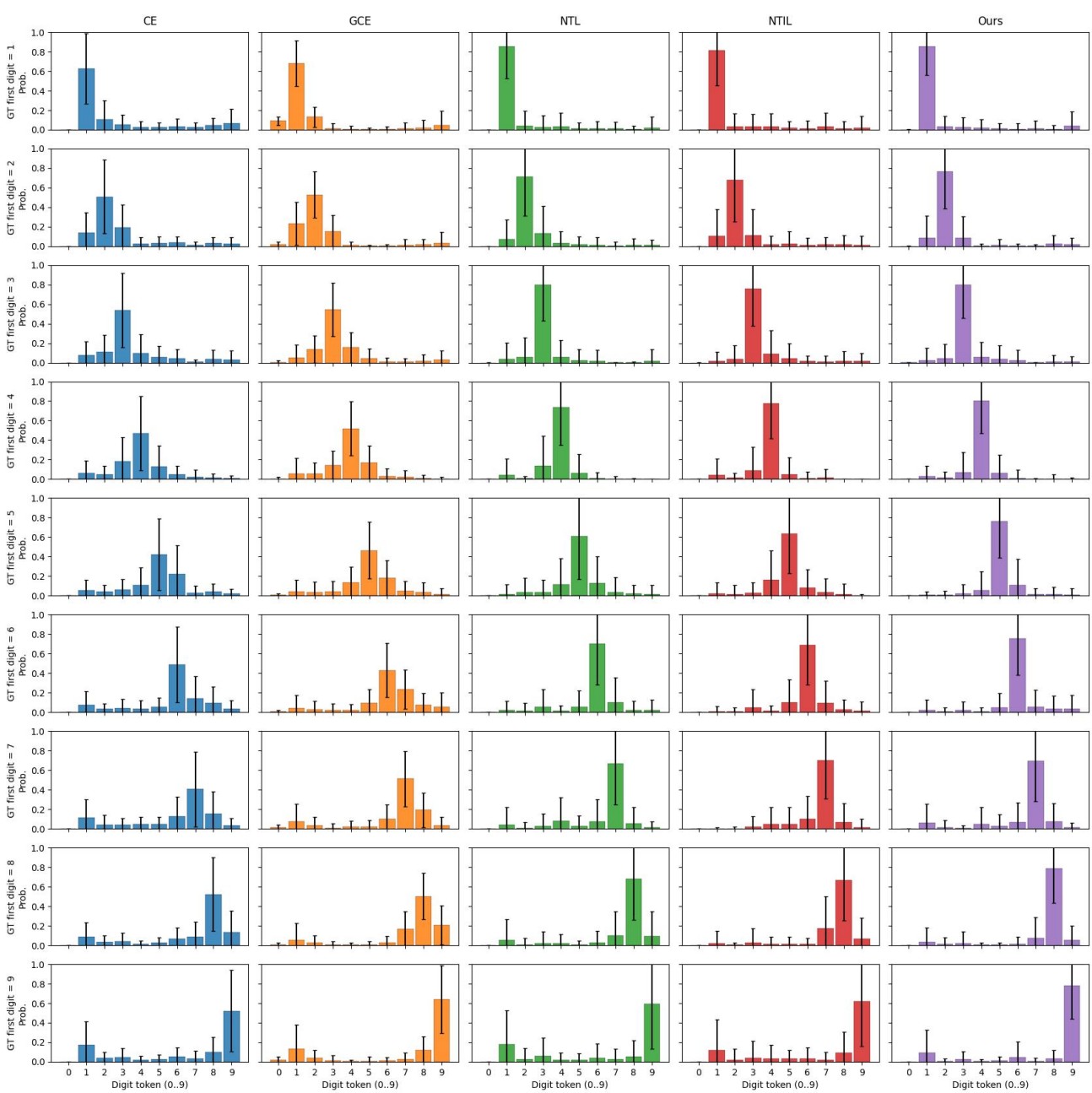

*Figure 6.* **Clock-Time digit distributions.** Rows bucket examples by the first digit of the ground-truth hour (1–9), and columns compare training objectives. Overall, our method yields the most concentrated distributions with the highest mass on the correct target digit across buckets.

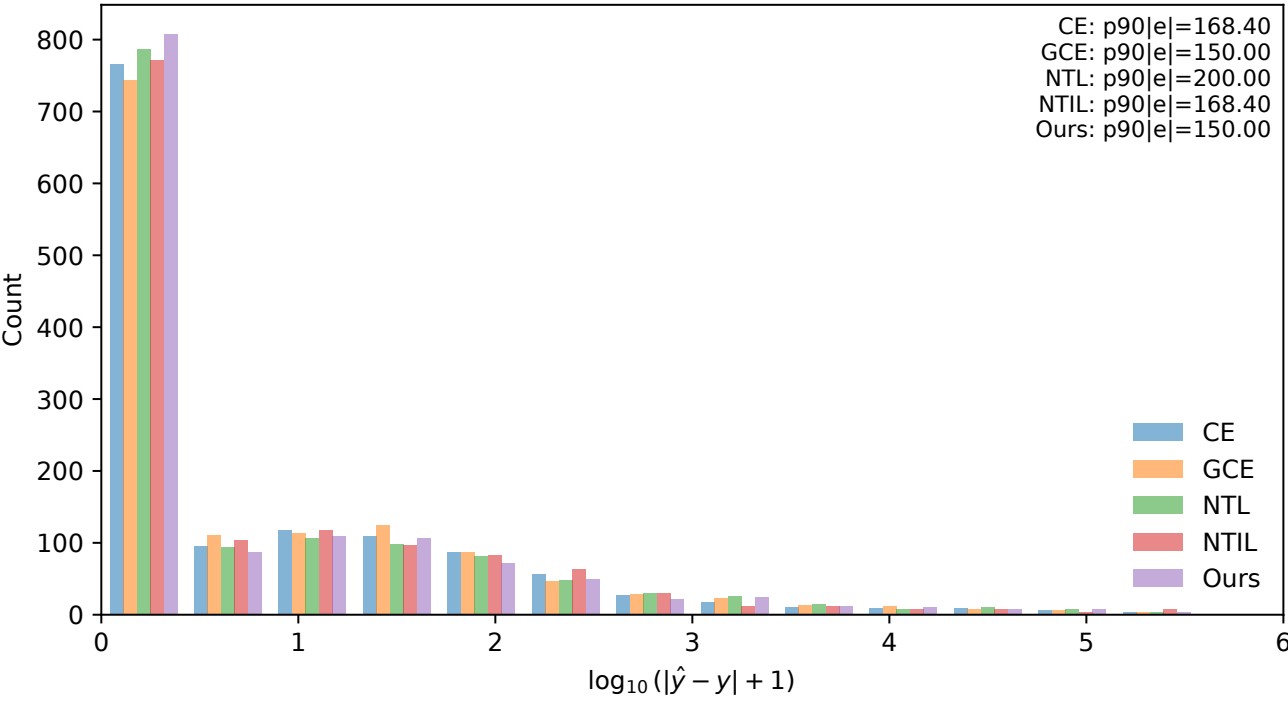

*Figure 7.* GSM8K overall error histogram on the common-valid subset, using $\log_{10}(|\hat{y} - y| + 1)$. Our distribution concentrates more on the left and exhibits a smaller tail (e.g., lower $p90(|e|)$), suggesting reduced large-magnitude errors.

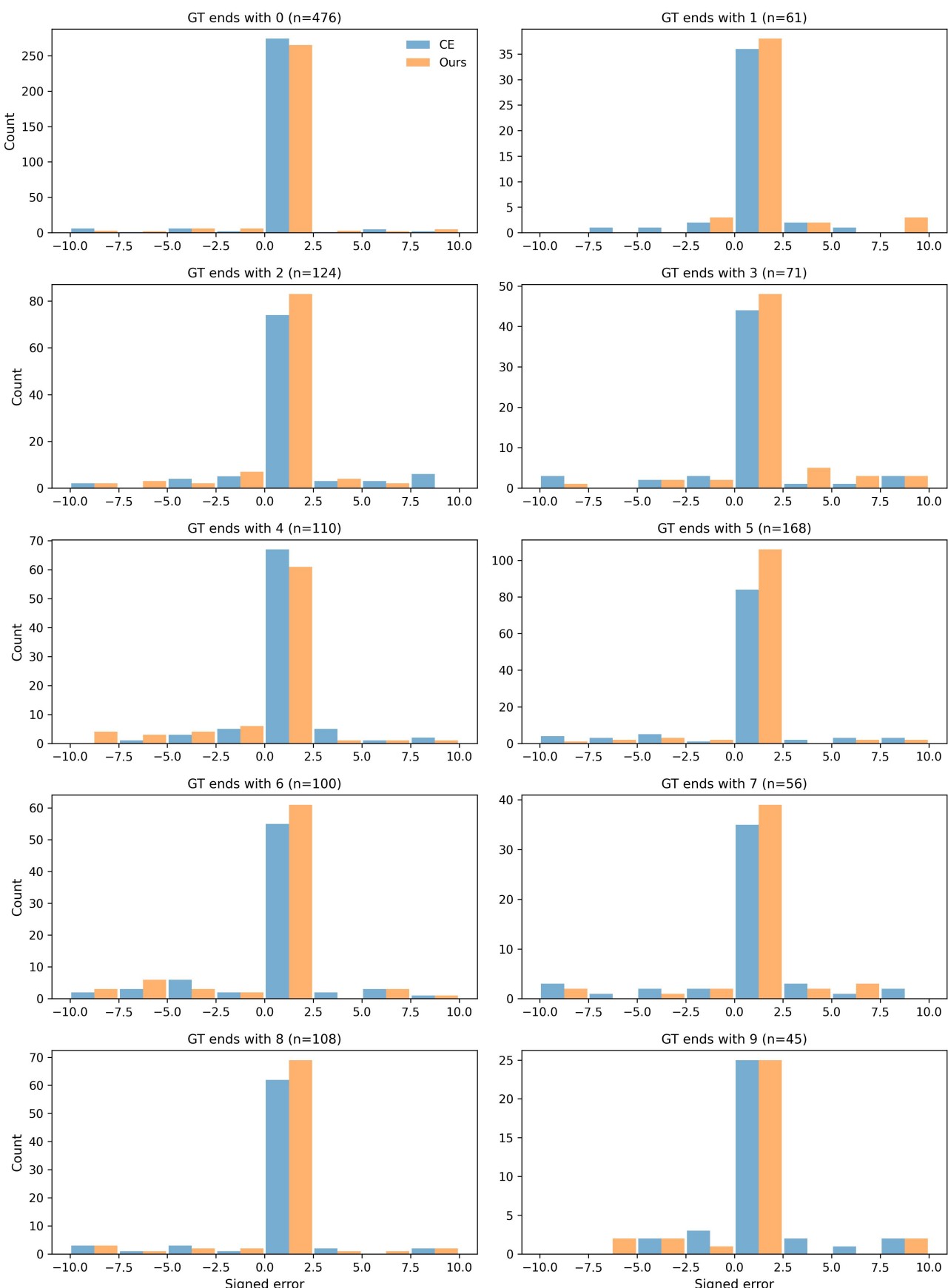

*Figure 8.* GSM8K signed error histograms $(\hat{y} - y)$ sliced by the last digit of the ground-truth answer. Each subplot overlays CE vs. Ours on the same subset. Overall, our errors concentrate more around zero for most digits, with mixed trends for boundary endings (0 and 9).

