# OpenReview forum: "Enhancing Numerical Prediction in LLMs via Smooth MMD Alignment"
_ICML.cc/2026/Conference — ICML 2026 regular_

### Official Review · Reviewer_cu6N · 2026-03-04

**Soundness:** 3
**Presentation:** 2
**Significance:** 3
**Originality:** 3
**Overall Recommendation:** 4
**Confidence:** 4

**Summary:**

Large Language Models (LLMs) usually suffer from precise numeric prediction. To address this challenge, this paper proposes SMMD, a novel training-time loss function, to improve the accuracy of LLMs in predicting numerical tokens. For positions where ground-truth labels are numeric tokens, SMMD restricts the probability distribution to a numeric sub-vocabulary, calculates an extra MMD loss based on the magnitude of the numerical differences ($L_{\text{MMD}}$), and enhances local consistency among close numerical values ​​through smoothing regularization ($L_{\text{smooth}}$).

Extensive experiments demonstrate that SMMD significantly improves the accuracy of numeric predictions on language-only LLMs and multimodal VLMs, across various numerically related tasks, and on models with different parameter sizes and architectures, outperforming several other loss functions. The authors also conducted detailed ablation experiments and hyperparameter sensitivity analyses.

**Compliance With Llm Reviewing Policy:**

Affirmed.

**Final Justification:**

The authors' rebuttal has addressed most of my concerns, but I think the important question regarding where SMMD is applied remains unresolved (even if considering the latest response of the authors). Therefore, I will maintain my overall score. But I will increase my confidence (3->4) because I am now more certain of my "weak accept" decision.

**Key Questions For Authors:**

1. Does the deterministic numeric parser $\text{parse}(\cdot)$ only perform a simple type conversion from string to number? Does it handle cases such as "two", "half", or "pi", where the string is not a straightforward numeric representation but still has a well-defined numeric value?

2. Is the sensitivity of the hyperparameters $\sigma$ and $\lambda$ related to $|\mathcal{V}_{\text{num}}|$ (the size of the numeric sub-vocabulary)?

3. Is SMMD loss more suitable for pre-training or Supervised Fine-Tuning (SFT)? Specifically, assuming a data sample is "Calculate 2+3. Answer: 5.", should we calculate the SMMD loss on '2', '3', and '5', or only on the final result '5'? What are the training settings in your experiments?

Clarifying the above question will help readers understand the paper and may change my evaluation of it.

**Limitations:**

Yes.

**Strengths And Weaknesses:**

### Strengths:
1. This paper has theoretical support and a comprehensive experimental design. It is also well-written and easy to understand.

2. This paper attempts to address an important problem regarding the accurate numerical prediction of LLMs. The proposed method demonstrates good performance and is easy to follow; thus, it may have an impact on future model training.

### Weaknesses:
1. This paper calculates the SMMD loss for **all** numeric tokens, which implicitly assumes that the closer they are to the ground truth, the better the numeric predictions are. But this is not always the case. The authors' method only works for regression tasks. The authors should discuss whether this method has a positive or negative impact on non-regression tasks (e.g., in code generation tasks where '2' in the function name `int2str` represents 'to'; in HTTP response status codes, where numbers only represent categories, etc.).

2. This article has some theoretical support, but lacks a clear motivation: why should we use kernel-based theory to improve LLM numerical prediction?

3. Figure 1 is not clear enough, especially subfigure (c). The authors do not explain how $\mu_p$ and $\mu_q$ are obtained. And in the lower half of subfigure (c), what do the circles connected by springs mean? Additionally, all the mathematical formulas in the figure are screenshots rather than selectable text, resulting in jagged edges and blurriness, which does not meet the standards of a high-quality academic paper.

---

> ### Author Rebuttal · Authors · 2026-03-30
>
> We appreciate the reviewer’s careful review and constructive comments on the paper. We address them below.
>
> **(1) Deterministic numeric parser.**
> Yes—the current parser is intentionally conservative. It therefore handles straightforward numeric strings, but **does not** currently map lexicalized forms such as `"two"` into numeric values. We chose this design to keep the construction of $V_{num}$ unambiguous. Extending it to richer semantic numerals is an interesting future direction.
>
> **(2) Where SMMD is applied.**
> All experiments in this paper are conducted in an **SFT** setting, not pretraining. However, the objective itself is compatible with **both** settings: it is defined token-locally on any supervised position whose ground-truth token lies in $V_{num}$.
>
> In our experiments, SMMD follows exactly the same supervision mask as CE: it is added on every supervised numeric target position, and is zero elsewhere. Thus, on SFT-style data, only the completion side receives SMMD. If the supervised completion contains intermediate numeric reasoning steps, then **all such numeric positions** are used—not just the final answer token.
>
> We also ran a minimal ablation on Qwen2.5-1.5B / GSM8K, comparing our default setting against an **answer-only** variant that applies SMMD only on final-answer positions:
>
> | Loss | Acc |
> |------|-----------|
> | CE | 54.99 |
> | SMMD\_answer\_mask | 55.27 |
> | SMMD | 57.77 |
>
> The answer-only variant is only slightly better than CE, while the default setting is clearly stronger. This suggests that applying SMMD to **all supervised numeric positions** is more effective than restricting it to the final answer alone, so we recommend the full setting as the default.
>
> To avoid ambiguity in examples such as “Calculate 2+3. Answer: 5.”: whether `2`, `3`, and `5` receive SMMD depends on the **training mask**. Under our SFT setup, only supervised answer-side tokens receive SMMD; in this toy example, that would typically mean only `5`. By contrast, if the same text appeared as pretraining data, all supervised numeric tokens in the sequence—including `2`, `3`, and `5`—would receive SMMD.
>
> **(3) Dependence on $|V_{num}|$ and hyperparameter sensitivity.**
> We agree that the preferred $\sigma$ and $\lambda$ may depend on the effective granularity of $V_{num}$, although we did not isolate this systematically in the current paper. More directly, what affects $\sigma$ is not just $|V_{num}|$, but the **scale of pairwise numeric distances**: with raw distances, moving from a digit-level vocabulary to a much larger integer vocabulary makes the same $\sigma$ effectively more local.
>
> In our implementation, we partially address this by **rescaling pairwise distances to a bounded range** (0 for identical values, approximately 1–9 otherwise) before constructing the kernel, so that the same $\sigma$ has a more comparable meaning across tokenizer settings. This does not fully remove dependence on $|V_{num}|$, but helps explain why the same default setting remains effective across different tokenizers ($|V_{num}|=10$ or $1000$) in our main experiments.
>
> **(4) Scope of applicability.**
> We agree that the current SMMD rule is deliberately simple: it activates whenever a **supervised** target token belongs to $V_{num}$. For the tasks studied here, this is an **appropriate bias**, because they are all numeric-target tasks where closeness in value is semantically meaningful. Our claim is therefore task-conditional rather than universal: we do not assume the same rule should be applied unchanged when numbers function only as symbols or category IDs (e.g., `int2str`, HTTP status codes), and we will make this scope boundary clearer in the revision. More broadly, we do not observe obvious degradation of general capability, with **MMLU** on Qwen2.5-1.5B remaining **61.32** for both CE and SMMD.
>
> **(5) Motivation for the kernel-based objective.**
> Our motivation is that the kernel view provides a natural way to turn **value distance** into a **PSD similarity structure** over the numeric sub-vocabulary. This gives us two useful properties simultaneously:
> 1. **global distribution alignment** via the MMD term $r^\top K r$, and
> 2. **explicit local smoothness control** via the Laplacian term $r^\top L r$.
>
> In other words, the kernel construction lets us move from raw numeric distances to a principled similarity graph, on which both alignment and local consistency become natural quadratic objectives.
>
> **(6) Figure 1 clarity.**
> We agree that Fig. 1 is not clear enough.  We will redraw it using proper vector text, explicitly clarify that $\mu_p$ and $\mu_q$ are the kernel mean embeddings used only as a conceptual illustration of the MMD term, and annotate that the circles and connections denote the nodes and similarity edges of the kernel-induced graph. We will improve the figure accordingly.
>
> We thank the reviewer again for these thoughtful comments and for helping us make the paper clearer and more precise.

---

> > ### Author Rebuttal · Reviewer_cu6N · 2026-04-03
> >
> > Thanks for the clarification and the additional experiments. Most of my concerns have been addressed. However, I still have one question regarding where SMMD is applied.
> >
> > Specifically, I wonder if the weaker performance of the **answer-only** variant might largely come from the **much smaller number of positions where the SMMD loss is applied**. In the answer-only setting, the loss can only be applied once per sample (i.e., on the final answer), whereas in the full setting it can be applied many more times across numeric tokens in intermediate steps. In that case, **the improvement might partly reflect more exposure to the SMMD training signal, rather than purely the choice of which positions receive SMMD**.
> >
> > Clarifying this distinction is crucial for the scalability of the proposed method. Since your additional experiment is just a minimal ablation on a relatively small dataset with a small model, the performance gain might heavily rely on the increased density of the loss signal. However, in **large-scale pretraining or SFT scenarios where models are trained to convergence** on massive datasets, the density of training signals is no longer the primary bottleneck. In such settings, the position of the SMMD loss will be insightful. Therefore, I think it would be highly beneficial to carefully isolate these two factors for better assessment of the potential of SMMD loss.

---

> > > ### Author Response · Authors · 2026-04-04
> > >
> > > We thank the reviewer for this helpful clarification. We agree that the current answer-only ablation does not fully disentangle where SMMD is applied from the resulting coverage of SMMD supervision.
> > >
> > > One implementation detail may help reduce this concern. In both our formulation and implementation, the SMMD term is **averaged** over all SMMD-active numeric positions within the batch, rather than summed, so the full variant is not simply benefiting from a mechanically larger loss due to having more active numeric tokens. We will revise the text accordingly and present this ablation with more appropriate scope.

---

### Official Review · Reviewer_as2V · 2026-03-12

**Soundness:** 4
**Presentation:** 3
**Significance:** 3
**Originality:** 3
**Overall Recommendation:** 5
**Confidence:** 4

**Summary:**

This paper introduces Smooth Maximum Mean Discrepancy (SMMD), a novel token-level loss function designed to improve the numerical perception of LLMs. Prior works has shown that CE ignores the metric distance between numbers, while recent optimal transport-based methods like NTL fail to constrain local smoothness, leading to unstable prediction distributions. SMMD addresses this by computing the MMD between predicted and target distributions, which is defined via a distance-induced kernel. Then SMMD regularizes the residual errors using a Dirichlet energy term on the graph Laplacian. The method demonstrates improvements on several numerical and arithmetic benchmarks compared to CE and NTL baselines.

**Compliance With Llm Reviewing Policy:**

Affirmed.

**Final Justification:**

After reading the authors' response and other reviewer' comments, I think the authors make a good rebuttal. Compared to prior numerical-based methods like NTL, SMMD is both more efficient and more precise. So I increase my score.

**Key Questions For Authors:**

- I am curious about whether training with SMMD negatively impact the model's performance on non-numeric benchmarks (e.g., standard text generation, MMLU).
- What is the running time complexity of SMMD compared to NTL-based baselines?

**Limitations:**

Yes. The authors mainly talk about the induced hyperparameters. I come across some potential future works of SMMD, including:

- Handling not digit-based tokenized LLMs (e.g., "12345" into "123" and "45" by T5 or [0-999] normalization by llama-3).
- Discovering whether such a better numerical cold start after finetuning (higher GSM8K score) leads to better post-trained checkpoint.

## References

[1] Teaching Metric Distance to Discrete Autoregressive Language Models. https://arxiv.org/abs/2503.02379

[2] FoNE: Precise Single-Token Number Embeddings via Fourier Features.  https://arxiv.org/abs/2502.09741

**Strengths And Weaknesses:**

## Strengths

- This paper is of soundness. The mathematical formulation is elegant and theoretically well-grounded. Modeling the numerical vocabulary as a graph and applying graph Laplacian regularization to the prediction residuals is a principled and interesting way for me. It is a good solution for the field which modifies CE to enhance the model's numerical ability.

- The provided code is clear and the paper is well-written.

## Weaknesses

- One flaw from my perspective is, the motivation behind SMMD is similar to DIST2 [1], a work that is not discussed by this paper. Both SMMD and DIST2 aim to inject a smooth metric distance into the token-level loss. While SMMD achieves this via Laplacian smoothing on prediction residuals within an RKHS, DIST2 directly softens the target distribution into a distance-aware Gaussian. Although the stories conveyed by these two paper are different, the intuition seems similar. I suggest the authors to discuss something on this.

- I don't know what's the advantage of modifying CE over other methods that also aim to promote the numerical ability. For example, a better numerical-aware position embedding can also be helpful [2]. Modifying the loss enables finetuning pretrained checkpoints for downstream tasks.

- The paper claims that NTL-based method demonstrates unsmooth distribution. I can intuitively understand that, but the provided visualization is not clear enough. I suggest adding more illustrative surrogate metrics for readers who are not familiar with this field.

---

> ### Author Rebuttal · Authors · 2026-03-29
>
> Thank you for the thoughtful and forward-looking review. We especially appreciate the pointers to DIST$^2$ and FoNE, as well as the suggestion to make the “smoothness” claim more quantitative. We address the points below.
>
> **(1) Relation to DIST$^2$.**
>
> We agree that DIST$^2$ and SMMD are motivated by a similar intuition: both inject metric distance into token-level supervision, but they differ substantially in mechanism. DIST$^2$ is closer to distance-aware target shaping (and thus conceptually closer to our GCE baseline), whereas SMMD performs kernel distribution matching with graph-Laplacian smoothing on the residual. In a best-effort reproduction, conducted without official code, DIST$^2$ reached 53.52 Acc on Qwen2.5-1.5B with GSM8K, below SMMD’s 57.77.
>
> **(2) Advantage of modifying the loss.**
>
> We agree that methods such as FoNE are valuable, but they address a different layer of the problem. FoNE operates at the **representation** level, while SMMD operates at the **objective** level. The practical advantage of SMMD is that it is **plug-and-play**: it does not modify any architecture and can be directly applied to existing pretrained checkpoints for downstream finetuning. In this sense, the two directions are largely **orthogonal** and potentially complementary.
>
> **(3) Quantifying local residual smoothness.**
>
> Our claim is that NTL-based objectives do **not** explicitly regularize local residual consistency along the numeric axis. Following your suggestion, we added **surrogate metrics** for the same Clock-Time first-digit setting as in Fig. 2. Let $r=p-q$ denote the residual between predicted $p$ and target $q$. In addition to Dirichlet energy, we report  **residual total variation (TV)** ($\sum_i |r_{i+1}-r_i|$) and **residual curvature** ($\sum_i |r_{i-1}-2r_i+r_{i+1}|$), which quantify local variation and oscillation across neighboring digits. As shown below, SMMD attains the **lowest** values on all three metrics, indicating smoother and more locally coherent residuals than NTL/NTIL.
>
> | Loss | First-digit Acc $\uparrow$ | Dirichlet energy $\downarrow$ | Residual TV $\downarrow$ | Residual curvature $\downarrow$ |
> |------|-----------------------------|-------------------------------|--------------------------|---------------------------------|
> | NTL  | 0.7507 | 1.5251 | 0.9086 | 1.5645 |
> | NTIL | 0.7389 | 1.5411 | 0.9471 | 1.6380 |
> | SMMD | 0.8181 | 1.0375 | 0.7045 | 1.2099 |
>
> **(4) Non-numeric benchmarks.**
>
> We evaluated **MMLU** (standard 5-shot via lm-eval) using the same Qwen2.5-1.5B checkpoints from our main GSM8K experiments. The overall average accuracy is the same (**61.32** for both CE and SMMD), and the category breakdown is nearly identical:
>
> | Method | STEM | Humanities | Social Sci. | Other | Average |
> |--------|------|------------|-------------|-------|---------|
> | CE   | $55.47 \pm 0.85$ | $54.94 \pm 0.68$ | $71.60 \pm 0.80$ | $66.72 \pm 0.82$ | $61.32 \pm 0.39$ |
> | Ours | $55.44 \pm 0.86$ | $54.86 \pm 0.68$ | $71.79 \pm 0.80$ | $66.69 \pm 0.82$ | $61.32 \pm 0.39$ |
>
> This is also consistent with our result on the SAMSum dataset in Appendix C.8, where SMMD remained very close to CE on dialogue summarization.
>
> **(5) Complexity and runtime.**
>
> In the one-hot-target setting, **NTL** reduces to an $O(N)$ distance-weighted sum over the numeric sub-vocabulary, whereas **SMMD** naively computes $r^\top K r$ and $r^\top L r$, i.e., $O(N^2)$ per numeric-target position. However, both $K$ and $L$ are **precomputed once**, and in practice $N$ is small (typically 10 or 1000). Empirically, the runtime overhead is modest. We measured the average per-step training time over the first 100 steps.
>
> **Qwen2.5-1.5B (digit-level tokenizer)**
>
> | Loss | Avg step time (ms) $\downarrow$ | Slowdown vs CE |
> |------|----------------------------------|----------------|
> | CE   | $322.29 \pm 2.97$ | $1.00\times$ (+0%) |
> | NTL  | $357.77 \pm 0.77$ | $1.11\times$ (+11.01%) |
> | NTIL | $929.07 \pm 3.19$ | $2.88\times$ (+188.27%) |
> | SMMD | $333.21 \pm 1.93$ | $1.03\times$ (+3.39%) |
>
> **SmolLM3-3B (multi-digit tokenizer)**
>
> | Loss | Avg step time (ms) $\downarrow$ | Slowdown vs CE |
> |------|----------------------------------|----------------|
> | CE   | $439.82 \pm 0.74$ | $1.00\times$ (+0%) |
> | NTL  | $468.98 \pm 2.92$ | $1.07\times$ (+6.63%) |
> | NTIL | $924.49 \pm 0.84$ | $2.10\times$ (+110.19%) |
> | SMMD | $449.40 \pm 0.48$ | $1.02\times$ (+2.18%) |
>
> Thus, while SMMD is formally higher-order than NTL in $N$, its **measured training overhead is only about 2–3% over CE** in both tokenizer regimes, and it remains far lighter than NTIL in practice.
>
> Finally, we agree that handling tokenization schemes beyond the digit-level setting is an important direction for future work. We are also exploring whether SMMD can provide a stronger numeracy-oriented cold start for the RL stage. We will include these as concrete future-work directions.
>
> Thank you again for your careful and patient review.

---

> > ### Author Rebuttal · Reviewer_as2V · 2026-04-01
> >
> > I appreciate the authors' response. My major concerns have been addressed. After carefully checking other reviewer' comments and the corresponding responses, I believe this paper would be of soundness and decide to increase my ratings.
> >
> > Additionally, though not so necessary, I suggest eloborating more on the necessity of numerical perceptional ability for modern LLM in the background part. Prevalent LLM agents can learn to understand numbers with tools (e.g., use Python sandbox to do calculation). Maybe the profits lie in comprehension in a paragraph with multiple numbers.

---

> > > ### Author Response · Authors · 2026-04-04
> > >
> > > We sincerely appreciate the reviewer’s positive feedback and supportive assessment. We are pleased that our response addressed the major concerns, and we will further improve the revised version, including a clearer discussion of the broader motivation and practical role of numerical perception in modern LLMs.

---

### Official Review · Reviewer_Loc5 · 2026-03-13

**Soundness:** 3
**Presentation:** 3
**Significance:** 3
**Originality:** 3
**Overall Recommendation:** 4
**Confidence:** 3

**Summary:**

This paper proposes Smooth Maximum Mean Discrepancy (SMMD) to improve numeric token prediction in autoregressive LLMs. The key idea is that the conventional cross-entropy loss treats numbers as unstructured categorical labels, ignoring metric relationships between values. SMMD introduces a value-distance-based kernel and smooth regularization to enforce the metric structure of numbers during learning, thereby improving prediction accuracy and local consistency.
The method is clean and promising, with consistent gains across multiple models and tasks. This is particularly relevant for scientific computing, financial analytics, and vision-alignment tasks that require highly accurate numerical outputs.

**Compliance With Llm Reviewing Policy:**

Affirmed.

**Key Questions For Authors:**

Could the authors clarify how robust SMMD is when nearby integers are decomposed into different token sequences, and whether the reported gains depend in part on favorable tokenization schemes rather than the objective itself?

A practical advantage claimed by the paper is that SMMD is lightweight and architectural-change-free, while NTIL is explicitly described as having substantially higher latency. However, the paper does not appear to report concrete per-step runtime or memory overhead for SMMD itself. Since deployability is important, could the authors provide measured training-time and memory overhead, ideally across both digit-level and larger numeric vocabularies?

The sensitivity plots suggest a meaningful dependence on these choices. Could the authors clarify how much tuning budget was used per task and baseline, and whether SMMD still maintains a consistent advantage under a strictly matched hyperparameter search protocol?

**Limitations:**

Although SMMD is designed to make token-level supervision value-aware, its notion of numerical proximity is still mediated by the tokenizer-defined numeric vocabulary. In practice, the paper notes that this vocabulary may range from digit tokens to larger sets that include multi-digit integers as single tokens, which means the effective distance structure can vary substantially across backbones. As a result, nearby numeric values may not always be represented in a way that faithfully reflects semantic closeness, especially when numbers are split into different token sequences.

While the paper argues that the kernel can be precomputed with negligible overhead, this discussion mainly concerns the kernel matrix itself and does not fully quantify the end-to-end training overhead of adding SMMD on top of cross-entropy. Since one of the practical appeals of the method is its architectural simplicity, explicit measurements of wall-clock training cost and GPU memory usage would be helpful to assess deployability more convincingly.

Some of the empirical gains may be task-dependent. The results are strong on mathematical reasoning and time-reading benchmarks, but the paper also notes that on ChartQA the objectives are broadly comparable and that performance there is often bottlenecked by visual grounding and value extraction rather than the numeric loss alone. This suggests that SMMD may be most beneficial when the dominant error source is numeric prediction itself, and less decisive when upstream perception or grounding errors dominate.

**Strengths And Weaknesses:**

Strengths
Clear motivation: The paper argues that standard cross-entropy does not encode metric meaning for numeric errors (how far off the prediction is), motivating value-aware supervision.
Broad empirical evaluation across backbones: The GSM8K/SVAMP table reports improvements for multiple model families and sizes, indicating strong scalability.

Weaknesses
1. Constraint by Tokenization: The method’s notion of numerical distance is constrained by the tokenizer; for instance, near-misses in multi-digit numbers may be over-penalized if the integers are split across different tokens.
2. Computational Latency: What is the measured per-step time increase (and GPU memory increase) when adding SMMD?
3. Hyperparameter Sensitivity: The proposed SMMD objective introduces additional hyperparameters, specifically the loss weight ($\lambda$) and kernel bandwidth ($\sigma$), which may require dataset-specific tuning to reach optimal performance.

---

> ### Author Rebuttal · Authors · 2026-03-29
>
> We appreciate the reviewer’s focus on practical robustness, deployability, and tuning fairness. These are exactly the issues that matter if a numeric-aware loss is to be practically useful, and we address them below.
>
> **(1) Dependence on tokenization.**
>
> SMMD operates at the token level, so the distance it uses is determined by the tokenizer’s numeric vocabulary. Its role is to improve supervision over the numeric structure already available under a fixed tokenizer, rather than to fully capture sequence-level numeral semantics.
>
> This token-level scope is also consistent with our empirical results: SMMD improves performance under both **digit-level** tokenizers (e.g., Qwen2.5 / Ministral, $|V_{num}|=10$) and tokenizers with many **multi-digit integer** tokens (e.g., SmolLM3 / Llama3, $|V_{num}|=1000$), suggesting that its gains are not tied to a single favorable tokenizer.
>
> **(2) Training-time overhead.**
>
> We agree that the paper should quantify end-to-end overhead more explicitly. We therefore measured training step time and peak allocated GPU memory under a controlled GSM8K setup on a single NVIDIA L20 GPU (batch size = 8, GA = 1). For each loss, we ran 120 training steps, discarded the first 20 warm-up steps, and reported the mean and std over the remaining steps, with 3 repeated runs. We report results for two representative backbones: Qwen2.5-1.5B with a digit-level tokenizer, and SmolLM3-3B with a multi-digit tokenizer.
>
> **Qwen2.5-1.5B (digit)**
>
> | Loss | Avg step time (ms) $\downarrow$ | Slowdown vs CE | Peak allocated (GB) |
> |------|----------------------------------|----------------|---------------------|
> | CE   | $322.29 \pm 2.97$ | $1.00\times$ (+0%) | $8.02$ |
> | NTL  | $357.77 \pm 0.77$ | $1.11\times$ (+11.01%) | $10.27$ |
> | NTIL | $929.07 \pm 3.19$ | $2.88\times$ (+188.27%) | $10.27$ |
> | SMMD | $333.21 \pm 1.93$ | $1.03\times$ (+3.39%) | $10.28$ |
>
> **SmolLM3-3B (multi-digit)**
>
> | Loss | Avg step time (ms) $\downarrow$ | Slowdown vs CE | Peak allocated (GB) |
> |------|----------------------------------|----------------|---------------------|
> | CE   | $439.82 \pm 0.74$ | $1.00\times$ (+0%) | $7.05$ |
> | NTL  | $468.98 \pm 2.92$ | $1.07\times$ (+6.63%) | $8.70$ |
> | NTIL | $924.49 \pm 0.84$ | $2.10\times$ (+110.19%) | $8.71$ |
> | SMMD | $449.40 \pm 0.48$ | $1.02\times$ (+2.18%) | $8.71$ |
>
> These results show that SMMD adds only **modest end-to-end overhead** relative to CE (about **2–3%** step-time increase across both tokenizer regimes), while remaining far lighter than NTIL in practice. This strengthens the deployability claim beyond the precomputed-kernel discussion in the paper.
>
> **(3) Tuning fairness and hyperparameter sensitivity.**
>
> We would like to clarify that our goal was **not** to heavily retune SMMD for each task or model. Instead, the main experiments use a **shared default setting** for SMMD: a single RBF kernel with $\sigma=2.0$ and $\lambda=3.0$ across experiments (Sec. 4.3), rather than aggressive per-task tuning.
>
> More broadly, the comparison protocol was kept aligned across methods: as described in Appendix C.3, all methods were evaluated under the same fine-tuning and evaluation pipeline.
>
> For the baselines, we likewise used consistent and non-arbitrary settings. Specifically:
> - **GCE:** we followed the original paper and used the same fixed recommended setting ($\sigma_{gce}=0.5$; Sec. 4.2) across all experiments.
> - **NTL:** we included a $\lambda$-scan in Fig. 5 on two representative settings (Qwen2.5-1.5B/GSM8K and Qwen2.5-VL-3B/Clock-Time), and adopted the best common setting ($\lambda=2.0$) in the main experiments.
> - **NTIL:** due to NTIL’s substantially higher training cost, we did not perform a separate hyperparameter search for it. Instead, we used the original implementation defaults for the sequence-level coefficients ($\alpha=\beta=\tau=0.2$) and set the overall loss weight to $\lambda=2.0$, following the best representative setting found for NTL, since both are fundamentally EMD-based objectives (Appendix C.2.2).
>
> In addition, the paper already reports a sensitivity analysis of SMMD with respect to both $\lambda$ and $\sigma$ (Fig. 3). This evidence is particularly relevant here, because it shows that SMMD’s gains are **not tied to a single cherry-picked hyperparameter choice**, but persist over a reasonably broad range of settings.
>
> Finally, we agree with your observation that SMMD is most beneficial when the dominant bottleneck is **numeric prediction itself**. This is consistent with our interpretation of the results: gains are strongest on math and clock-time tasks, while on ChartQA, upstream visual grounding can dominate, leaving less room for the numeric loss alone to help. We will clarify this scope boundary more explicitly in the revision.

---

> > ### Author Rebuttal · Reviewer_Loc5 · 2026-04-03
> >
> > My comments and concerns were well solved by authors, thanks for your response. I have no further questions.

---

> > > ### Author Response · Authors · 2026-04-04
> > >
> > > We appreciate the reviewer’s positive feedback and thoughtful assessment. We are glad that our response addressed the concerns, and we will further improve the presentation and clarification in the revised version.

---

### Official Review · Reviewer_zmUb · 2026-03-15

**Soundness:** 3
**Presentation:** 3
**Significance:** 3
**Originality:** 3
**Overall Recommendation:** 4
**Confidence:** 2

**Summary:**

The authors propose a method for improving numerical predictions in LLMs via smooth MMD (SMMD) alignment, which builds on MMD by incorporating value-distance kernels over numeric tokens and graph-based smoothness. They evaluate this method across multiple LLMs and VLM backbones on four numeric-target tasks of mathematical reasoning, arithmetic calculation, clock-time recognition, and chart question-answering. They show that SMMD consistently improves numerical predictions across language-only and vision-language tasks. They also perform various ablations to verify where the gains are coming from. Finally, they show that SMMD is robust under simple default settings, while the hyper-parameters can vary across datasets.

**Compliance With Llm Reviewing Policy:**

Affirmed.

**Final Justification:**

My main concerns are resolved and I maintain my original positive score.

**Key Questions For Authors:**

Please refer to weaknesses.

**Limitations:**

yes

**Strengths And Weaknesses:**

Strengths:

- This paper addresses an important issue of mismatch between the metric structure of numeric values and the training signal used to model them.
- The authors claim that their approach is the first one to use kernel distribution matching to supervise numeric token prediction.
- The method is "plug-and-play," requiring no architectural changes and small computational overhead during training.

Weaknesses:

- As mentioned by the authors in the conclusion section, the notion of the distance is constrained by the tokenizer and the method is limited by how the model breaks down numbers.
- On the complex reasoning tasks, the performance gains from SMMD are relatively small, and the paper does not report standard deviations over multiple training runs or perform statistical significance testing, which makes it hard to know how significant the improvements actually are.
- The evaluation focuses exclusively on numeric-target tasks, and the authors do not report results on standard language benchmarks. Therefore, it remains unclear whether the observed gains in numeracy come at the expense of linguistic fluency or broader world knowledge.

---

> ### Author Rebuttal · Authors · 2026-03-29
>
> We thank the reviewer for the careful reading and constructive comments. We address them below.
>
> **(1) Tokenizer limitation.**
>
> We agree with this point and also acknowledge it in the paper’s limitation section. SMMD is defined over token-level numeric units, so its notion of distance is determined from the tokenizer. We do not claim that SMMD fully resolves sequence-level numeral semantics. Rather, it is designed to improve supervision **under the token-level numeric setting** used by current autoregressive LMs.
>
> Importantly, the gains are not tied to a single tokenizer type. In our main results, SMMD improves performance under both **digit-level tokenizers** (e.g., Qwen2.5 / Ministral, where $|V_{num}|=10$) and tokenizers with many **multi-digit integer tokens** (e.g., SmolLM3 / Llama3, where $|V_{num}|=1000$). We will clarify this scope more explicitly in the revision and leave sequence-level extensions to future work.
>
> **(2) Multi-run results and statistical significance.**
>
> We agree that reporting only single-run results was insufficient. However, due to the high computational cost and time requirements of LLM training, we used **GSM8K**, our main reasoning dataset, as a representative setting and repeated training **3 times per method** for the main LLM backbones. SMMD remains the best-performing method in mean accuracy on all tested models:
>
> | Model        | CE              | GCE             | NTL             | NTIL            | Ours            |
> |--------------|-----------------|-----------------|-----------------|-----------------|-----------------|
> | Qwen2.5-0.5B | $30.32 \pm 0.75$ | $28.93 \pm 0.80$ | $30.15 \pm 0.18$ | $30.45 \pm 0.29$ | $\mathbf{31.49 \pm 0.25}$ |
> | Qwen2.5-1.5B | $54.67 \pm 0.60$ | $52.71 \pm 0.27$ | $55.87 \pm 0.34$ | $55.22 \pm 0.57$ | $\mathbf{57.77 \pm 0.38}$ |
> | SmolLM3-3B   | $67.65 \pm 0.96$ | $66.77 \pm 0.23$ | $64.08 \pm 0.37$ | $69.42 \pm 0.65$ | $\mathbf{70.28 \pm 0.46}$ |
>
>
> As an additional check, we conducted **paired t-tests** on the results from three seed-matched runs. SMMD shows statistically significant improvements over both CE and the strongest prior baseline on representative backbones:
>
> | Model | Comparison | p-value |
> |-------|------------|---------|
> | Qwen2.5-0.5B | Ours vs CE | 0.0653 |
> | Qwen2.5-0.5B | Ours vs NTIL | 0.0688 |
> | Qwen2.5-1.5B | Ours vs CE | 0.0034 |
> | Qwen2.5-1.5B | Ours vs NTL | 0.0082 |
> | SmolLM3-3B | Ours vs CE | 0.0280 |
> | SmolLM3-3B | Ours vs NTIL | 0.0209 |
>
> Thus, the gains are statistically significant on Qwen2.5-1.5B and SmolLM3-3B, both against CE and against the strongest prior baseline for each model. This is also consistent with the broader pattern in our main result, where SMMD improves performance across multiple datasets and backbones. On the smaller Qwen2.5-0.5B, the trend remains positive in mean accuracy but does not reach significance under this limited-sample test.
>
> **(3) Evaluation on general benchmarks.**
>
> We agree that numeracy gains should not come at the expense of general capability. To check this, we evaluated the GSM8K-trained checkpoints on MMLU (standard 5-shot via lm-eval), reporting accuracy (Acc, %). For Qwen2.5-1.5B, the overall average is **unchanged** (61.32 for both CE and SMMD), and the category breakdown is nearly identical:
>
> | Method | STEM | Humanities | Social Sci. | Other | Average |
> |--------|------|------------|-------------|-------|---------|
> | CE   | $55.47 \pm 0.85$ | $54.94 \pm 0.68$ | $71.60 \pm 0.80$ | $66.72 \pm 0.82$ | $61.32 \pm 0.39$ |
> | Ours | $55.44 \pm 0.86$ | $54.86 \pm 0.68$ | $71.79 \pm 0.80$ | $66.69 \pm 0.82$ | $61.32 \pm 0.39$ |
>
>
> This suggests that SMMD improves numerical prediction **without sacrificing broader general capability**. In addition, Appendix C.8 already includes a sanity check on the SAMSum dataset, where SMMD remains very close to CE on dialogue summarization, further supporting its compatibility with standard text modeling objectives.
>
> We appreciate these suggestions and will revise the paper to clarify the method scope, the multi-run statistics, and the broader evaluation of general capability.

---

> > ### Author Rebuttal · Reviewer_zmUb · 2026-04-04
> >
> > Thanks for your response. My concerns are mainly resolved.

---

> > > ### Author Response · Authors · 2026-04-04
> > >
> > > We thank the reviewer for the acknowledgment and positive feedback, and are glad that our clarifications helped address the concerns.

---

### Decision · Program_Chairs · 2026-04-30

**Decision:**

Accept (regular)

**Comment:**

This paper introduces Smooth Maximum Mean Discrepancy (SMMD), a novel token-level loss function designed to improve numerical perception and prediction in autoregressive language models. The proposed approach addresses the limitations of conventional cross-entropy loss, which treats numbers as unstructured categorical labels and ignores the metric distances between values. SMMD computes the discrepancy between predicted and target distributions using a distance-induced kernel and regularizes residual errors with a Dirichlet energy term on the graph Laplacian to ensure local consistency.

The authors conducted extensive evaluations across multiple LLM and VLM backbones on various numerically related tasks. The experimental results demonstrate consistent improvements across both language and vision-language domains. SMMD also proved to be more efficient and precise than prior numerical-based baselines like NTL.


The reviewer consensus is uniformly positive. The reviewers highlighted that this method is clean, promising. During the rebuttal period, the authors successfully addressed most of the concerns raised by the reviewers. Following the rebuttal, one reviewer explicitly raised their score due to the method's proven efficiency and precision compared to existing work. Another reviewer maintained their positive evaluation and increased their confidence score. Given the robust empirical evidence, clear methodological contribution, and overall positive reviewer feedback, the paper merits acceptance.